# How Has Online Digital Technology Influenced the On-Site Visitation Behavior of Tourists during the COVID-19 Pandemic? A Case Study of Online Digital Art Exhibitions in China

Yanqing Xia [1,2] 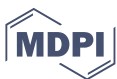

1 College of Art and Design, Zhejiang Guangsha Vocational and Technical University of Construction, Dongyang 322103, China; xyq520@pukyong.ac.kr
2 Industrial Design Department, Pukyong University, Busan 48513, Republic of Korea

**Abstract:** The COVID-19 pandemic has had a significant impact on the global tourism industry, leading to a decrease in peoples' willingness to travel and a sense of insecurity regarding tourist destinations. Therefore, restoring people's willingness to travel is the greatest challenge faced by this industry in the post-pandemic era. The tourism industry requires innovative solutions to achieve sustainable recovery. While there is a considerable amount of research on its recovery during the pandemic, there are few studies exploring people's willingness to travel to encourage sustainable and resilient recovery in the post-pandemic era. This study employed a quality model to examine the satisfaction and intention of tourists towards the application of online digital art exhibitions under the influence of COVID-19. The aim was to investigate the promoting role of online digital art exhibitions in the sustainability and resilient recovery of the tourism industry. To achieve these objectives, this study focuses on the online digital art exhibition of Song Dynasty figure paintings launched by China Central Television (CCTV), with post-exhibition surveys conducted and 512 valid questionnaires collected. The research model and hypotheses are tested using structural equation modeling. The results of this study indicate that travelers' intentions to engage in on-site visits through online digital exhibitions are determined by three factors: perceived value, satisfaction, and art therapy. Furthermore, online digital art exhibitions not only represented the most important form of tourism during the pandemic, but they also provided significant psychological healing. They have become a driving force for the transformation of the current culture and tourism industry and the promotion of its sustainable development. This research provides a benchmark for future research on the tourism industry, and it offers new research directions in the field of sustainable tourism.

**Keywords:** sustainable tourism; tourism industry; COVID-19; behavioral intention; online digital exhibition

## 1. Introduction

The tourism industry provides livelihoods to millions of people and enables billions to appreciate diverse cultures and the natural world. In some countries, tourism accounts for more than 20% of the gross domestic product (GDP), making it the third-largest export sector in the global economy [1]. However, the COVID-19 pandemic has severely impacted the tourism industry, affecting economies, livelihoods, and public services across continents and bringing the industry to a temporary halt [2]. According to forecasts by the World Travel and Tourism Council (WTTC), the COVID-19 pandemic was expected to cause a loss in global tourism to the value of USD 2.1 trillion in 2020 and jeopardize 75 million tourism jobs [3]. Rebuilding market confidence, revitalizing the tourism economy, and achieving high-quality development of the tourism industry have become focal points in academia and industry post-pandemic.

The outbreak of the pandemic has posed unprecedented challenges to the tourism industry, yet it has also presented an opportunity for transformation and change. The

emergence of online digital technologies has facilitated the digital transformation and innovation of the tourism sector [4]. Digital cultural tourism initiatives have emerged during the pandemic, leveraging new-generation technologies such as virtual reality (VR), augmented reality (AR), and artificial intelligence (AI) to create immersive experiences. These initiatives include virtual reality-based scenic spots, entertainment, and museums, and other new forms of culture and tourism experience, fostering new consumer behavior and reshaping people's travel preferences and tourism concepts [5]. Emphasizing safety in tourism and the ability to explore the world from home, and prioritizing user experience, have become paramount [5]. Under the circumstances of the pandemic, online digital technologies have gained more attention than ever from tourists and destination organizations. Online virtual tours of museums and exhibitions and other forms of "cloud tourism" have become the new norm since the pandemic, stimulating people's interest in tourism. Concurrently, online digital technologies are reshaping the manner in which customers plan their travel and search for destination information. By leveraging digitized environments, these technologies enable customers to experience products, services, or locations prior to their physical visit [6]. The increasing reliance on and desire for such online digital environments reflects a gradual shift in tourism perspectives, with individuals predominantly opting for online virtual tourism. Consequently, online digital technologies have opened up new avenues for the revitalization and recovery of the tourism industry, providing innovative approaches to travel and tourism experiences.

Previous studies have explored the use of on-site digital technologies [7] and VR technology for enhancing tourists' travel experiences [8], as well as the impact of VR technology on intentions to make repeat visits to tourist destinations [9,10]. However, the perspectives explored in these studies have been relatively narrow, focusing solely on the impact of digital technologies on tourism destinations. They predominantly concentrate on the "technology" itself, without delving into the influence of online digital technologies on "people" after the pandemic. In particular, the shift in individuals' tourism perspectives and their heightened concerns for personal well-being and health have significantly dampened their willingness to travel. Yet, the revival of tourist demand is considered a key factor in stimulating the recovery of the tourism industry [11]. Consequently, there is an urgent need to address how to effectively guide tourists in regaining their willingness to travel. In fact, tourists are highly interested in online digital technologies and seek to understand how these platforms can provide humanistic care [12,13]. However, there have been few attempts to evaluate user satisfaction with online digital technology applications and their subsequent impact on travel intentions. How do online digital technologies influence tourists' on-site visitation behavior during the COVID-19 pandemic? What are the underlying factors affecting this behavior? Do they contribute to stimulating tourists' travel intentions and promoting the recovery and sustainable development of the tourism industry in the post-pandemic era?

To address the aforementioned issues, this study introduces a comprehensive research model that incorporates factors derived from perceived quality (content, system, and personalized service) and expectation confirmation. From the perspective of tourists' expectation confirmation, the research model explores the effects of perceived quality of online digital art exhibitions, user expectation confirmation, perceived value, perceived enjoyment, and art therapy on satisfaction with online digital art exhibitions and on-site visitation. By examining how online digital technology influences the physical visit behavior of tourists during the COVID-19 pandemic, this research can assist museum curators and other cultural tourism institutions in leveraging digital technology to increase public engagement with art, promote the digital transformation of the culture and tourism industry, stimulate innovation in this sector, unleash the multiplicative effects of digitization on the industry, and facilitate its sustainable development.

## 2. Research Theory and Hypothesis

### 2.1. COVID-19 and Online Digital Art Exhibitions

In the spring of 2020, as the novel coronavirus began to spread globally, the true meaning of globalization seemed to resonate with people, as the impact of the virus extended beyond trade disruptions and stagnant tourism to endanger the art world [14]. As a result of the pandemic, planned art fairs and major exhibitions in various countries were hindered, while museums in many countries remained closed, presenting challenges in cross-border artistic collaboration [15]. To cater to audiences who were unable to attend in person, many museums swiftly transitioned their exhibitions to online platforms, offering "virtual exhibitions" for remote viewing. Suddenly, being "online" became the hottest topic of discussion. It is during this distinctive period that the rapid development of digital art took place.

Online digital art exhibitions refer to the integration of online digital technologies with humanistic art, combining rational thinking and artistic sensibility. These exhibitions are founded on the development of online digital technologies and encompass a fusion of artistic expression and human perception [16]. Online exhibitions, as a form of exhibition accessible anytime and from anywhere through the internet on computers and smartphones, represent one of the most effective ways to disseminate digital information in any field [17]. Online digital art exhibitions represent a new artistic language that blends the contemporary era of new media with traditional art. Leveraging technology for human–computer interaction as a medium, these exhibitions enhance visitors' perceptual experiences and create a multidimensional and dynamic interactive environment.

### 2.2. Perceived Quality

The concept of perceived system quality was introduced by DeLone and McLean [17], and it is generally defined as "the extent to which users perceive the performance of a system". Subsequently, this model was updated by Delone and McLean [18], who proposed three types of information system quality—system quality, information quality, and service quality, also known as perceived quality. Perceived quality can be defined as customers' judgment of the overall excellence or superiority of a product. These three indicators, evaluated based on individual perceptions, are considered primary predictors of perceived quality [19]. Currently, the Information System Success Model has been successfully applied in the field of e-commerce. This model explains users' adoption of various information systems, for instance, websites, online shopping, mobile applications, online learning, and online travel through VR augmented reality [20–24].

Content quality refers to indicators associated with the content of an e-commerce website, including the relevance, completeness, and comprehensibility of the provided information. Delone and McLean [17] emphasized the importance and relevance of content quality within the Information System Success Model, with various studies further emphasizing its significance. System quality refers to "a system in which the expected characteristics of mobile devices and web browsing services are considered available for user use" [25]. Delone and McLean [17] demonstrated that system quality strongly influences the success of information systems, with indicators relating to system quality and overall system performance measured based on factors such as usability, reliability, functionality, and performance. Service quality is defined as the overall judgment of or attitude towards the quality of a service, and it refers to consumers' overall impression of its advantages and/or disadvantages. Zhao et al. [26] identified service quality as an important determinant of information system effectiveness.

In this study, content quality refers to the organization of information provided in terms of its accuracy, relevance, and personalization within the context of online digital exhibitions on cultural heritage; system quality pertains to the degree of security, speed, reliability, and convenience of the webpages and mobile interfaces; and good service quality is characterized by visually appealing graphics and immersive interactive experiences.

Online digital art exhibitions represent innovative information technology primarily based on new media. Although research on the perceived quality of online digital exhibitions is limited in terms of user expectations confirmation, studies have extensively explored perceived quality and expectation confirmation in various domains, such as online restaurant reviews, online shopping, and online library resources. Joo and Choi [27] confirmed that users' perceptions of the content quality of online library resources positively influence the confirmation of their expectations. Park [28] added that the confirmation of users' expectations is positively influenced by system and service quality in the context of smart wearable devices. Therefore, a similar trend is likely to apply to users of online digital art exhibitions. Based on this premise, we propose the following hypotheses:

**H1a**: There exists a positive correlation between the content quality of online digital art exhibitions and visitor expectation confirmation.

**H1b**: There exists a positive correlation between the service quality of online digital art exhibitions and visitor expectation confirmation.

**H1c**: There exists a positive correlation between the system quality of online digital art exhibitions and visitor expectation confirmation.

### 2.3. The Expectation Confirmation Model

The Expectation Confirmation Model (ECM), proposed by the renowned American scholar Bhattacherjee [29], is widely recognized as a prominent theory for explaining the post-adoption behavior of users [30]. Initially, it was developed to elucidate the relationships between factors influencing the repeat purchase behavior of consumers. It posits that there exists an inherent relationship between the "expectations" prior to product purchase, the "perceived performance" after purchase, the "degree of confirmation" between prior expectations and perceived performance, "satisfaction" after purchase, and consumers' "intention to repurchase." These factors were identified to be interconnected. In 2001, Bhattacherjee introduced the Expectation Confirmation Model, which comprises four variables: expectation confirmation, perceived usefulness, satisfaction, and continued usage intention.

Expectation confirmation as a key factor in the ECM that plays a significant role in determining users' perceived satisfaction with information systems and services, which, in turn, influences their system usage. Furthermore, as satisfaction is typically defined as a user's overall evaluation of their experience with a specific information system or service, Mason and Nassivera [31] identify satisfaction as one of the core determinants of behavioral intention. Therefore, this study incorporates expectation confirmation, satisfaction, and behavioral intention from the ECM into its research model.

Additionally, Hsu and Lin [32] demonstrate that perceived value replaces perceived usefulness. Users engage with online digital art exhibitions for various reasons, such as leisure, enjoyment, and learning, rather than to achieve specific goals or enhance performance. Previous research has shown that in the tourism domain, the primary factors influencing behavioral intention are perceived quality, perceived value, and satisfaction [33,34]. Thus, perceived value deserves greater attention.

#### 2.3.1. Confirmation

Bhattacherjee defines expectation confirmation as the degree to which information system users' pre-usage expectations are confirmed after system usage. In the context of this study, expectation confirmation refers to a user's evaluation of the degree to which their overall perception of the online digital art exhibition of cultural heritage aligns with their pre-visit expectations. In the Expectation Confirmation Model, expectation confirmation is a key factor that directly influences user satisfaction with system usage. The higher the confirmation of users' prior expectations, the greater their perceived satisfaction with the usage experience. For instance, Vena-Oya et al. [35] found that the confirmation of high expectations significantly influences people's satisfaction with tourist destinations.

Chen et al. [36] demonstrated that expectation confirmation plays a decisive role in determining satisfaction with online shopping. Based on these findings, the following hypothesis is proposed:

**H2**: There is a positive correlation between expectation confirmation and visitors' satisfaction with online digital art exhibitions.

Previous research has shown a strong and important relationship between expectation confirmation and perceived value. Lin et al. [37] revealed that higher expectation confirmation in IPTV leads to greater perceived value. Research found that confirmation of users' expectations of mobile services plays a decisive role in their perception of value and satisfaction [38]. Therefore, the following hypothesis is proposed:

**H3**: There is a positive correlation between expectation confirmation and visitors' perceptions of the value of online digital art exhibitions.

Furthermore, Halilovic and Cicic [39] introduced an extended Expectation Confirmation Model that integrates expectation confirmation with the affective structure of individual users, specifically, the perception of enjoyment. This integration further complements the continuity of perceived information technology quality. Research has shown that expectation confirmation has a positive impact on the perception of enjoyment. Based on this, we propose the following hypothesis:

**H4**: There is a positive correlation between user expectation confirmation and visitors' perceptions of enjoyment in online digital art exhibitions.

### 2.3.2. Satisfaction

Satisfaction, as defined by Oliver [40], represents the discrepancy between consumers' prior expectations and their perception of the product's performance after consumption. Both the ECM and subsequent studies have confirmed that satisfaction is a primary determinant of users' behavioral intentions. Visitors feel satisfied when their experience meets or exceeds their expectations. Numerous studies have established that user satisfaction is an important indicator of subsequent behavioral intentions [41,42], and that satisfaction with online travel experiences has a positive impact on users' behavioral intentions [42–44]. The more satisfied users are with online travel, the stronger their intention to engage in actual visits. Therefore, the following hypothesis is proposed:

**H5**: There is a positive correlation between satisfaction and visitors' intentions to engage in on-site visitation behavior.

### 2.3.3. Perceived Value

Perceived value refers to the overall evaluation of the utility of a product or service [45]. Woodruff [46] considers a product's performance, attributes, and usage effects to be customer value factors that influence the attainment of customer goals. Murphy et al. [47] point out that perceived value, in the context of tourism, refers to the comparison made by tourists between the quality of their tourism experiences and the resources (e.g., money, time) invested in their travel. In this study, perceived value refers to users' overall evaluation of the cultural value, aesthetic value, and service value experienced during the expectation confirmation phase of the digital online exhibition of cultural heritage. Perceived value is relative in nature, as it is comparative, personal, and contextual, and is inherently prioritized, affective, and cognitively emotional [48].

Numerous studies have demonstrated a causal relationship between perceived value and satisfaction, effectively explaining tourist satisfaction and destination choice [49]. Gallarza and Gil Saura [50] argue that perceived value is a prerequisite to experiencing satisfaction in a tourism context. Wei et al. [51] identify tourists' perception of value after engaging in cultural heritage tourism as one of the most significant factors influencing their

satisfaction with a destination and their intention to revisit. Chen and Chen [52] find that visitors' perception of value in heritage tourism has a positive impact on their satisfaction. Based on this, we can hypothesize the following:

**H6**: There is a positive correlation between the perceived value of online digital art exhibitions and visitor satisfaction.

A substantial body of empirical research suggests that perceived value may be a stronger predictor of user behavior compared to satisfaction or quality. For example, Cronin et al. [53] examine the relationships between perceived value, satisfaction, and behavioral intentions across six different service industries, and find that perceived value directly influences customer satisfaction and behavioral intentions in all industries except healthcare. Pura [54] analyzes the direct impact of perceived value on behavioral intentions in a service context. The results of their study indicate that there is a significant influence of perceived value on behavioral intentions, with higher perceived value leading to stronger behavioral intentions. Based on this, we can hypothesize the following:

**H7**: The perceived value of online digital art exhibitions has a positive impact on visitors' intentions to engage in on-site visitation behavior.

### 2.3.4. Perceived Enjoyment

Perceived enjoyment refers to the extent to which users perceive the activity of using a specific information system as enjoyable [55]. Thong et al. [56] were the first to investigate the relationship between perceived enjoyment and the continued intention to use mobile internet services. By extending perceived enjoyment to the Extended Confirmation Model, a better understanding of the continued behavior in mobile internet services can be achieved. Perceived enjoyment represents the level of enjoyment experienced by users during the usage process, independent of the usage outcome. In the context of this research, perceived enjoyment primarily refers to the pleasant emotions experienced by users after engaging in online digital exhibitions on cultural heritage, with greater emphasis on fully immersive experiences through virtual interaction with art exhibitions.

The concept of perceived enjoyment has been widely studied across different domains. In healthcare environments, visual art has been found to alter patients' emotions, evoking a sense of pleasure in hospital rooms and providing patients with a sense of mental well-being [57]. Ulrich [58] emphasizes that natural environments elicit pleasant emotions and psychological responses, emphasizing therapeutic perception based on emotions. Day [59] suggests that the full appreciation or enjoyment of artwork can serve as a healing pathway for individuals. In their book, Malchioldi and Lippin [60] state that scientific evidence supports the positive impact of all forms of therapeutic art, because they combine sensibility, emotions, and cognition, allowing individuals to fully enjoy and experience life. Chen [61] discusses how audiences easily find presence after interacting with high-quality interactive art, which provides fundamental sensory enjoyment and awakens emotional resonance in individuals, thus serving as a form of therapy and restoration. Based on these observations, we can propose the following hypothesis:

**H8**: The perceived enjoyment of visitors towards online digital art exhibitions is positively correlated with their perception of art therapy.

### 2.4. Artistic Healing

The origin of "art therapy" can be traced back to Hans Prinzhorn, who published "Artistry of the Mentally Ill" in 1922, where he suggests that the configuration impulse in the drawings of psychiatric patients can be traced back to the universal psychological history of humankind. This study also marked the beginning of the intersection between psychology and the field of art [62]. With the dissemination of art and the development of art therapy, in the 1940s, psychiatrist Margaret Naumburg solidified the concept of "art therapy". Si-

multaneously, artist Adrian Hill and art teacher Edith Kleinman recognized the therapeutic potential of art and the creative process as an intervention for psychological issues. Under the guidance of art therapists, patients enhance their self-awareness, and experience personal growth and therapeutic effects. Over time, art therapy has evolved into a diverse field, encompassing various modalities such as film, music, drama, games, performance, visual arts, and sandplay. Art therapy is no longer solely targeted toward psychiatric patients and has also become a means of healing for individuals seeking to overcome psychological challenges, engage in self-exploration, and express themselves artistically.

In this study, art therapy primarily refers to the "visual arts as therapy" approach, where users primarily view online digital art exhibitions to experience aesthetic pleasure, emotional release, and emotional resonance with the artworks and to attain a state of relaxation, thereby reducing mental anxiety and fostering a sense of social belonging.

Previous research has shown limited discussion on satisfaction and behavioral intentions regarding art therapy and online digital art exhibitions. However, scholars have recognized the positive effects of art therapy in settings such as museums and healthcare [63]. A study by Harris et al. [64] indicated that art therapy in hospital environments is a key indicator of hospital satisfaction. This viewpoint has been supported by subsequent studies [65,66]. Zhang [67] demonstrated the significant impact of decorative art in creating emotionally therapeutic landscapes in public spaces and satisfying individuals' spiritual pursuits. Additionally, Zhang et al. [68] argued that perceived therapeutic effects have a positive influence on the loyalty and behavioral intentions of visitors in urban parks. Kreitler and Kreitler [69] assessed the impact of the art therapy process on individuals' behavioral intentions. Based on these findings, we can propose the following hypotheses:

**H9**: There is a positive correlation between art therapy and visitor satisfaction with online digital exhibitions.

**H10**: There is a positive correlation between art therapy and visitors' on-site visitation behavior.

### 2.5. Research Model

Figure 1 presents a conceptual model developed based on the proposed hypotheses.

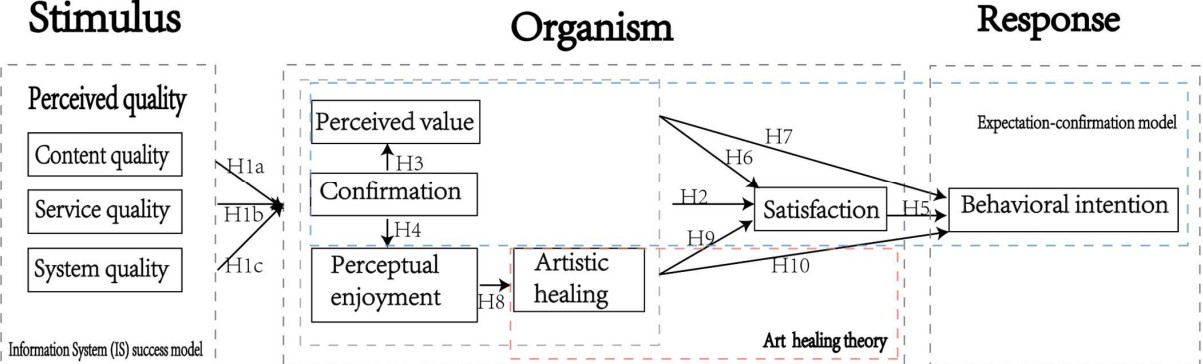

**Figure 1.** Research model.

The conceptual model presented in this study redefines the Stimulus–Organism–Response (S-O-R) theory by incorporating nine variables: system quality, information quality, service quality, expectation confirmation, perceived value, perceived enjoyment, art therapy, visitor satisfaction, and visitor intention to revisit. The S-O-R theory aims to explain users' behavioral responses to external stimuli. In fact, online digital exhibitions inherently operate within the S-O-R framework, as they provide users with visual aesthetics and services, serving as a distinctive form of psychological stimulation. This service (or stimulus) leads to emotional changes and cognitive enhancement in exhibition visitors, which, in turn, influence their final behavioral decisions, specifically regarding their intention to seek offline exhibition experiences (R). Despite the aforementioned logical

relationships, there is a lack of research specifically focused on online digital exhibitions, and their application in this context remains largely unexplored. To address our research inquiries, we propose ten hypotheses (Figure 1).

## 3. Research Context and Study Methodology

### 3.1. Song Dynasty Figures Online Digital Exhibition

This study focuses on an online digital art exhibition titled "Millennium Tone: Portraits of Song Dynasty Figures", which was launched by CGTN, the China Global Television Network, in January 2023. As noted by the renowned scholar Chen Yinke, "The culture of the Chinese nation has evolved over thousands of years, reaching its pinnacle during the Song Dynasty" [70]. This highlights the significance of Song Dynasty culture in the development of Chinese culture. This special digital exhibition features a curated selection of over 200 precious artworks from more than 10 museums worldwide, showcasing the painted figures of the Song Dynasty to an audience of thousands. The exhibition can be accessed via both computers and mobile devices, and it employs the cutting-edge Unity engine, developed for the gaming industry, to create interactive 3D scenes of ancient art. This innovative approach brings the thousand-year-old characters and scenes depicted in Song Dynasty paintings to life, transcending the static nature of traditional artworks. The exhibition website consists of six main thematic digital halls, namely, "Companions", "Elegant Demeanor", "Officials", "Performers", "Commoners", and "Immortals" (Figure 2). These halls present the various figures depicted in Song Dynasty paintings, ranging from children and maidservants to scholars, artists, merchants, and Daoist immortals. By offering a close-up experience in which visitors can view the depicted characters, the exhibition provides insights into everyday life in the Song Dynasty, including clothing, cosmetics, and entertainment. Each exhibition area provides vivid interpretations from different perspectives, ensuring a dynamic and interactive experience. Visitors can enjoy and explore 110 high-definition Song Dynasty paintings, whereby the ancient artworks to come alive and engage the audience in an immersive and enjoyable manner (Figure 3).

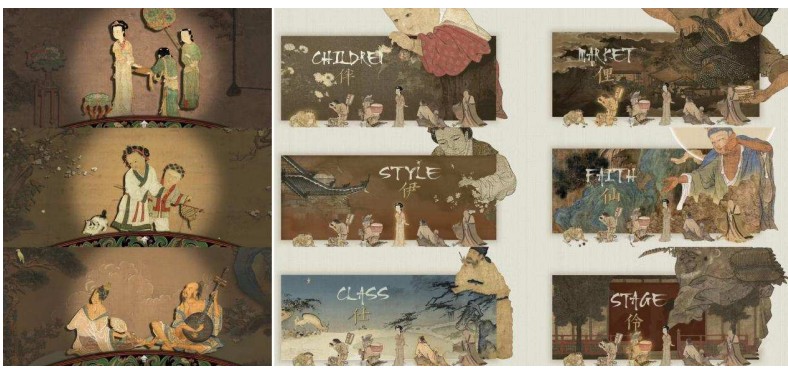

**Figure 2.** Six main thematic digital exhibition halls.

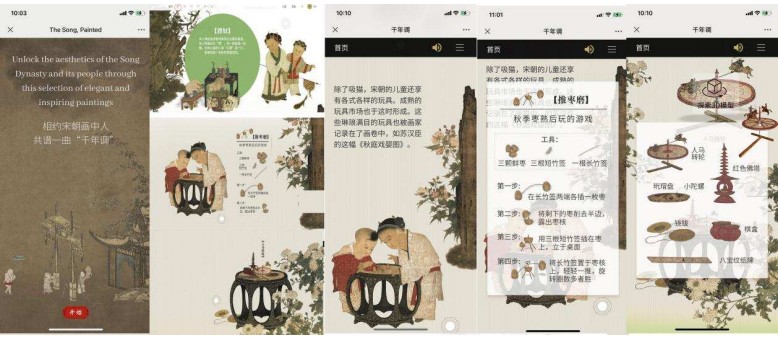

**Figure 3.** Exhibition content.

### 3.2. Questionnaire Design and Measurements

After designing the measurement indicators for each potential variable, this study used existing scales from previous research to create the initial survey questionnaire on factors influencing offline visits by users of the online digital art exhibition. The survey questionnaire consisted of two parts: (1) basic information on the survey participants (gender, age, occupation, etc.), and (2) 36 measurement items for 9 potential variables. To ensure the validity of the final survey data, the research team conducted a preliminary investigation of the initial questionnaire before the formal survey, collecting 126 test data for analysis; moreover, the research team optimized the questionnaire by deleting five items with insufficient reliability and low factor loading, and adding two items on whether users are interested in Song Dynasty culture and whether they would like to visit offline museums. The final questionnaire consisted of 31 measurement items, as shown in Table 1.

**Table 1.** Measurement of the variables.

| Constructs | | Items | Explanations | Source(s) |
|---|---|---|---|---|
| Perceived quality | Content quality | QC1 | Online art exhibition content provided aesthetic information of figure painting in Song Dynasty. | [6,19] |
| | | QC2 | Online art exhibition content was easy to understand. | |
| | | QC3 | Online art exhibition content was very rich. | |
| | Service quality | QS1 | In my opinion, the color scheme of this online art exhibition was very beautiful. | |
| | | QS2 | Online art exhibition incorporated animation effects. | |
| | | QS3 | Online art exhibition screen was high definition. | |
| | | QS4 | Online art exhibition had visually appealing visuals. | |
| | System quality | SQ1 | Online art exhibition software was easy for me to operate. | |
| | | SQ2 | Online art exhibition software was very user-friendly for me. | |
| | | SQ3 | Online art exhibition software was very safe for me. | |
| Confirmation | | CF1 | The experience of using this online art exhibition was better than I expected. | [29] |
| | | CF2 | Online art exhibition provided me with better services than I expected. | |
| | | CF3 | The overall visual effect of the online art exhibition was better than I expected. | |
| | | CF4 | Overall, my expectations for using online art exhibitions had mostly been met. | |
| Perceptual value | | PU1 | Appreciating online art exhibitions made me feel very literary value. | [29] |
| | | PU2 | It was very convenient for me to enjoy such a beautiful digital art exhibition online. | |
| | | PU3 | Overall, enjoying online art exhibitions enhanced my aesthetics. | |
| Perceptual enjoyment | | PE1 | When I was immersed in the exhibition of cultural paintings from the Song Dynasty, I did not realize the passage of time. | [29] |
| | | PI2 | The introduction of children's toys in the Song Dynasty provided me with fun. | |
| | | PI3 | The use of online art exhibitions had sparked my interest in Song Dynasty culture. | |
| Satisfaction | | SF1 | I was very satisfied with the overall experience brought by online digital art exhibition. | [29] |
| | | SF2 | Online art exhibition had brought me physical and mental relaxation, and I feel very satisfied. | |
| | | SF3 | Overall, I was very satisfied with this online digital art exhibition. | |
| Artistic healing | | AH1 | Seeing such harmonious timbre and figure painting of Song Dynasty could help me forget my troubles for a while. | [71,72] |
| | | AH2 | Seeing the Song Dynasty's "Lotus Pavilion Baby Play" made me understand the love of parents for their children from ancient times to the present, bringing me a sense of happiness. | |
| | | AH3 | Appreciating the online art exhibition of the Song Dynasty inexplicably gave me a sense of confidence, which comes from Chinese culture. | |
| | | AH4 | Online art exhibition gave me a breathing space during a busy day, allowing me to relax both physically and mentally. | |
| Continuing to Access Behavior | | CB1 | I wanted to go to the museum to enjoy the authentic figure painting of the Song Dynasty. | [29] |
| | | CB2 | I was happy to recommend it to my friends for appreciation, and if there was a chance, I would also go to the museum with my friends or family to have a look. | |
| | | CB3 | Overall, I would try to visit more offline museums in the future to experience firsthand experiences. | |
| | | CB4 | Currently, offline museums also hold such exhibitions, and I really want to go and experience them on-site. | |

### 3.3. Data Collection

This study was conducted following the launch of the Chinese digital exhibition "Online Guochao Digital Special Exhibition" on CGTN in January 2023. The researchers recruited 542 participants from across China who had not previously visited the exhibition. Participants ranged in age from 18 to 60 years old and were asked to view the entire exhibition before completing the questionnaire. Informed consent was obtained from all participants, and a random sampling technique was used. A seven-point Likert scale (ranging from 1 (strongly disagree) to 7 (strongly agree)) was utilized. The study data were analyzed using SPSS 27.0 and AMOS 27.0, including Confirmatory Factor Analysis (CFA) and Structural Equation Modeling (SEM). A total of 532 responses were collected from the on-site survey and subjected to encoding analysis. Responses with missing or consistently identical answers were excluded, resulting in 512 valid questionnaires with an effective response rate of 96.24%. Table 2 presents the demographic information of the survey respondents.

**Table 2.** Respondents' demographic information.

| Measure | Items | Frequency ($n$ = 512) | Percentage |
|---|---|---|---|
| Gender | Male | 241 | 47.1% |
| | Female | 271 | 52.9% |
| Age | Under 20 years old | 41 | 8% |
| | 20~29 | 197 | 38.5% |
| | 30~39 | 162 | 31.6% |
| | 40~49 | 84 | 16.4% |
| | Above 50 years old | 28 | 5.5% |
| Education | Junior college | 123 | 24% |
| | Bachelor's degree | 249 | 48.6% |
| | Master's degree or above | 140 | 27.3% |
| Occupation | Student (high school, college, graduate, etc.) | 142 | 27.7% |
| | Clerk | 165 | 32.2% |
| | Personnel (teacher, lawyer, doctor, civil servant, etc.) | 139 | 27.1% |
| | Professional | 52 | 10.2% |
| | Other | 14 | 2.7% |
| | Very uninterested | 9 | 1.8% |
| Have you ever been interested in Song Dynasty culture before? | Uninterested | 61 | 11.9% |
| | Commonly | 228 | 44.5% |
| | Interested | 139 | 27.1% |
| | Very interested | 75 | 14.6% |
| | Very dislike | 19 | 3.7% |
| | Dislike | 162 | 31.6% |
| Did you enjoy going to museums and exhibitions before? | Commonly | 261 | 51% |
| | Like | 40 | 7.8% |
| | Very like | 30 | 5.9% |

## 4. Results

### 4.1. Common Method Deviation

The data for this study were collected from a single source (participants or respondents), and the survey method involved self-perception and self-reporting. This data collection method is susceptible to common method bias, which may arise from the shared measurement environment, contextual factors, and the characteristics of the items themselves. To mitigate the artificial covariance between the predictor and criterion variables caused by the same measurement environment, contextual factors, and item characteristics, the study adopted the approach proposed by Cham et al. [73]. During the data collection phase, the research team intentionally designed the questionnaire items for different variables on separate pages, allowing respondents to have sufficient rest between pages and reducing the common method variance resulting from using the same scale. Additionally, the research team conducted Harman's single-factor test and exploratory factor analysis

(EFA) to examine the presence of common method bias. A total of 31 items were loaded together for the EFA, and the results showed a Kaiser–Meyer–Olkin value of 0.910, with a significance level of $p < 0.05$, indicating the suitability of the questionnaire data for factor analysis. The EFA revealed the presence of nine factors with eigenvalues greater than 1. Although the first unrotated principal component accounted for 34.25% of the variance, which is less than the recommended threshold of 40%, it suggests that common method bias did not have a significant impact on this study.

### 4.2. Descriptive Analysis

The descriptive statistics of the constructs employed in the research model are summarized in Table 3.

**Table 3.** Descriptive information of the constructs used in the research model.

| Construct | Mean (Standard Deviation) | Construct | Mean (Standard Deviation) |
|---|---|---|---|
| Content quality | 5.739 (1.084) | Perceptual enjoyment | 5.275 (1.408) |
| Service quality | 5.644 (1.096) | Satisfaction | 5.378 (1.314) |
| System quality | 5.308 (1.358) | Artistic healing | 5.298 (1.182) |
| Confirmation | 5.444 (1.083) | Behavioral intention | 5.636 (1.026) |
| Perceptual value | 5.753 (1.140) | | |

### 4.3. Measurement Model

To assess the measurement reliability and validity of our sample data, we conducted a confirmatory factor analysis (CFA) on all the item variables in our model. Convergent validity measures whether an item can effectively reflect its corresponding variable, while discriminant validity measures whether two variables have statistically significant differences [74]. Table 4 presents the values of the scale, items, standardized loadings, average variance extracted (AVE), composite reliability (CR), and Cronbach's alpha.

**Table 4.** Descriptive information of the constructs used in the research model.

| Factors | Items | Cronbach's Alpha | Factor Loading | AVE | CR |
|---|---|---|---|---|---|
| Content quality | QC1<br>QC2<br>QC3 | 0.871 | 0.813<br>0.880<br>0.807 | 0.696 | 0.873 |
| Service quality | QS1<br>QS2<br>QS3<br>QS4 | 0.912 | 0.863<br>0.857<br>0.845<br>0.832 | 0.721 | 0.912 |
| System quality | SQ1<br>SQ2<br>SQ3 | 0.900 | 0.863<br>0.893<br>0.844 | 0.752 | 0.901 |
| Confirmation | CF1<br>CF2<br>CF3<br>CF4 | 0.902 | 0.863<br>0.847<br>0.849<br>0.784 | 0.699 | 0.903 |
| Artistic healing | AH1<br>AH2<br>AH3<br>AH4 | 0.913 | 0.823<br>0.855<br>0.876<br>0.850 | 0.725 | 0.913 |
| Perceptual value | PU1<br>PU2<br>PU3 | 0.898 | 0.860<br>0.884<br>0.847 | 0.746 | 0.898 |
| Perceptual enjoyment | PE1<br>PE2<br>PE3 | 0.914 | 0.861<br>0.906<br>0.882 | 0.780 | 0.914 |
| Satisfaction | SF1<br>SF2<br>SF3 | 0.930 | 0.900<br>0.903<br>0.907 | 0.816 | 0.930 |
| Behavioral intention | CB1<br>CB2<br>CB3<br>CB4 | 0.895 | 0.805<br>0.837<br>0.851<br>0.804 | 0.680 | 0.895 |

As shown in Table 4, all the standardized factor loadings exceed 0.7, and the t-values indicate that these factor loadings are significant at the 0.001 level. The composite reliabilities (CR) of all the variables are greater than 0.7, and the average variance extracted (AVE) is greater than 0.5 for all variables. Therefore, all the indicators in this study are higher than the standard values, indicating good convergent validity of the measurement scale [75]. Moreover, the Cronbach's alpha coefficients for each item exceed the value of 0.70 recommended by Hundleby, indicating that the scale also has good reliability [76].

Based on the analysis results presented in Table 5, it is evident that in this study's discriminant validity test, the standardized correlation coefficients between all dimensions are smaller than the square roots of the corresponding AVE values. This observation indicates that there is good discriminant validity among all the dimensions in this study.

**Table 5.** Correlation matrices and discriminant validity.

| | 1 | 2 | 3 | 4 | 5 | 6 | 7 | 8 | 9 |
|---|---|---|---|---|---|---|---|---|---|
| Behavioral intention | **0.824** | | | | | | | | |
| Satisfaction | 0.420 | **0.903** | | | | | | | |
| Perceptual enjoyment | 0.486 | 0.400 | **0.883** | | | | | | |
| Perceptual value | 0.483 | 0.361 | 0.348 | **0.864** | | | | | |
| Artistic healing | 0.639 | 0.412 | 0.452 | 0.431 | **0.851** | | | | |
| Confirmation | 0.429 | 0.393 | 0.477 | 0.337 | 0.374 | **0.836** | | | |
| System quality | 0.421 | 0.255 | 0.371 | 0.286 | 0.365 | 0.45 | **0.867** | | |
| Service quality | 0.392 | 0.327 | 0.296 | 0.438 | 0.367 | 0.374 | 0.329 | **0.849** | |
| Content quality | 0.359 | 0.246 | 0.296 | 0.289 | 0.238 | 0.39 | 0.364 | 0.300 | **0.834** |

Note: The items on the diagonal on bold represent the square roots of the AVE.

The analysis results in Table 5 clearly demonstrate that in the context of discriminant validity testing in this study, the standardized correlation coefficients between each pair of dimensions are all smaller than the square root of the average variance extracted (AVE) values corresponding to those dimensions. This finding indicates that there is good discriminant validity among the dimensions in this study.

*4.4. Structural Model*

4.4.1. Fit Indices

The fit indices for both the research model and the measurement model were computed. As presented in Table 6, these indices were considered generally acceptable based on the recommendations from previous studies.

**Table 6.** Fit indices.

| Fit Indices | chi2/df | GFI | AGFI | CFI | TLI | RMSEA |
|---|---|---|---|---|---|---|
| Recommended values | <300 | >0.900 | >0.800 | >0.900 | >0.900 | <0.080 |
| Measurement model | 1.330 | 0.936 | 0.920 | 0.988 | 0.986 | 0.025 |
| Research model | 1.783 | 0.913 | 0.896 | 0.971 | 0.968 | 0.039 |

As demonstrated in Table 6, all the fit indices obtained in this study surpass the recommended values, indicating that the research model employed exhibits a satisfactory fit.

4.4.2. Hypothesis Tests

The path coefficients and their significance are summarized in Figure 4 and Table 7. As depicted in the figure, all 10 hypotheses were tested and supported at a significance level of 0.05. The results of this study demonstrate that users' intentions to engage in offline visits are primarily predicted by three variables: art therapy (H10, β = 0.314, $p < 0.001$), perceived value (H7, β = 0.508, $p < 0.001$), and satisfaction with enjoyment (H5, β = 0.141, $p < 0.001$). These variables collectively explain 48% of the variance in users' intentions to engage in offline visits.

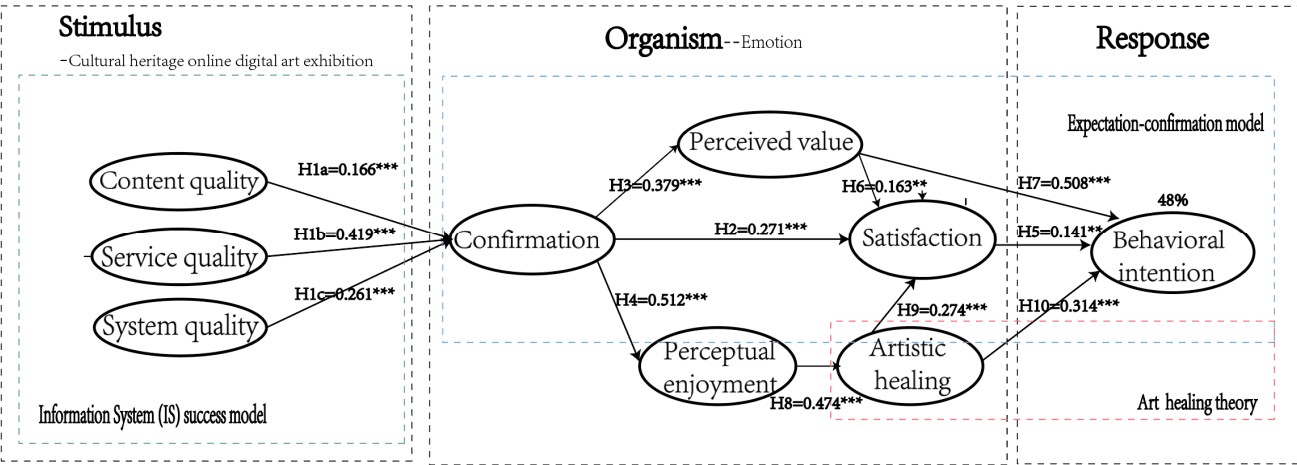

**Figure 4.** The research model. (Note: ** *p* <0.01; *** *p* <0.001).

**Table 7.** Fit indices and results of hypotheses test.

| Hypothesis | Estimate | S.E. | C.R. | Results |
|---|---|---|---|---|
| H1a: Content quality → Confirmation | 0.166 | 0.049 | 4.615 *** | Supported |
| H1b: Service quality → Confirmation | 0.419 | 0.044 | 4.894 *** | Supported |
| H1c: System quality → Confirmation | 0.261 | 0.038 | 6.441 *** | Supported |
| H2: Confirmation → Satisfaction | 0.271 | 0.069 | 3.827 *** | Supported |
| H3: Confirmation → Perceptual value | 0.379 | 0.051 | 7.813 *** | Supported |
| H4: Confirmation → Perceptual enjoyment | 0.512 | 0.060 | 10.956 *** | Supported |
| H5: Satisfaction → Behavioral intention | 0.141 | 0.033 | 3.264 ** | Supported |
| H6: Perceptual value → Satisfaction | 0.163 | 0.052 | 2.901 ** | Supported |
| H7: Perceptual value → Behavioral intention | 0.508 | 0.038 | 11.165 *** | Supported |
| H8: Perceptual enjoyment → Artistic healing | 0.474 | 0.038 | 10.027 *** | Supported |
| H9: Artistic healing → Satisfaction | 0.274 | 0.050 | 5.391 *** | Supported |
| H10: Artistic healing → Behavioral intention | 0.314 | 0.036 | 7.267 *** | Supported |

(Note: ** *p* < 0.01; *** *p* < 0.001).

In the Emotional Organism, perceived value (H6, β = 0.250, *p* < 0.001), art therapy (H9, β = 0.515, *p* < 0.001), and expectation confirmation (H2, β = 0.271, *p* < 0.001) positively influence satisfaction.

Perceived quality (H1a, β = 0.166, *p* < 0.001), service quality (H1, β = 0.419, *p* < 0.001), and system quality (H1c, β = 0.261, *p* < 0.001) all have positive effects on expectation confirmation. Additionally, expectation confirmation has a positive impact on perceived value (H3, β = 0.250, *p* < 0.001) and satisfaction with enjoyment (H4, β = 0.474, *p* < 0.001).

The results of the mediation analysis are presented in Table 8. In addition to the three variables with direct effects on behavioral intention, user expectation confirmation and satisfaction have two indirect effects on users' intentions to engage in offline visits. (1) Indirect effect 1 (0.072, 95% confidence interval does not include 0): Expectation confirmation → Perceived value → Satisfaction (significant mediation effect). (2) Indirect effect 2 (0.071, 95% confidence interval does not include 0): Expectation confirmation → Perceived value → Art therapy → Satisfaction (significant mediation effect), accounting for 16.6% and 16.5% of the total effect, respectively, and acting as partial mediators.

**Table 8.** Indirect effects on intention to engage in offline visits.

| Intermediate Path | Point Estimation | Product of Coef | | Bootstrapping | | | | Proportion |
|---|---|---|---|---|---|---|---|---|
| | | | | Bias-Corrected 95%CI | | Percentile 95%CI | | |
| | | SE | z | Lower | Upper | Lower | Upper | |
| Total effects | 0.393 | 0.069 | 5.696 | 0.330 | 0.589 | 0.325 | 0.585 | |
| CF → PV → SF | 0.072 | 0.030 | 2.400 | 0.024 | 0.146 | 0.018 | 0.135 | 16.6% |
| CF → PE → AH → SF | 0.071 | 0.021 | 3.381 | 0.038 | 0.124 | 0.034 | 0.119 | 16.5% |
| CF → SF | 0.316 | 0.075 | 4.213 | 0.185 | 0.474 | 0.177 | 0.458 | 68.9% |

## 5. Results and Discussion

The global tourism industry has been severely impacted by the COVID-19 pandemic, which is ongoing and becoming the new norm. As the world gradually recovers, stakeholders in the tourism industry are seeking policies that can facilitate its revival in this new context. Therefore, one of the primary objectives of this study is to assess the impact of online digital art exhibitions on the recovery and revitalization of the tourism industry. This study analyzes the decisive factors in visitors' on-site visitation behavior as influenced by the widespread adoption of online digital technologies during the COVID-19 pandemic. The findings, presented in Table 5, provide support for all ten of our proposed hypotheses.

This study demonstrates that the three dimensions of perceived quality in online digital art exhibitions (content quality, system quality, and service quality) are crucial in confirming visitors' expectations, with service quality having the highest impact coefficient. These findings are consistent with those of previous studies [77–79]. Specifically, the more stimuli visitors experience from online digital art exhibitions on cultural heritage, the more positive their confirmation becomes. The iconic representation of Song Dynasty figure paintings in Chinese aesthetics serves as an attraction for visitors, as the perceived visual aesthetic appeal and diverse interactive experiences stimulate higher levels of expectation confirmation. Therefore, as people's travel demands evolve, future online curators should consider leveraging the increased cognitive recognition of this type of exhibition tourism, focusing on broader sharing, more effective interactions, richer sensory experiences, and convenient information dissemination.

Secondly, the empirical results of this study reveal the existence of two mediating effects between expectation confirmation and satisfaction, which indirectly influence visitors' intentions to visit offline museums. These effects include the mediation of perceived value and enjoyment, as well as enjoyment and art therapy. This implies that perceived value, enjoyment, and art therapy are important pathways through which expectation confirmation influences satisfaction. Expectation confirmation serves as a significant motivating factor for satisfaction, and while previous studies have confirmed the impact of expectation confirmation on satisfaction [80,81], the internal process mechanism underlying the influence of expectation confirmation on satisfaction remains unclear. This study discovers that perceived value, enjoyment, and art therapy can further elucidate the relationship between expectation confirmation and satisfaction. Perceived value plays a partially mediating role in confirmation and satisfaction, which is consistent with previous research demonstrating the important link between perceived value and expectation confirmation and satisfaction [82]. Additionally, for the first time, this study reveals that expectation confirmation influences satisfaction through the mediation of enjoyment and art therapy, with remote mediation effects being significant. In other words, the higher the expectation confirmation of visitors to online cultural heritage art exhibitions, the higher their perceived enjoyment, and the greater their perception of art therapy, leading to physical relaxation and mental comfort; this, in turn, affects visitors' satisfaction with online cultural heritage digital art exhibitions. These findings indicate that the importance of expectation confirmation in ensuring satisfaction can also be understood through the mediating roles of enjoyment and art therapy.

Thirdly, this study reveals that perceived value, satisfaction, and art therapy, as latent variables, exert a positive and significant influence on visitors' offline visitation behavior.

These findings align with those of previous research [83,84], confirming the consistent impact of perceived value and satisfaction on visitors' offline visitation behavior. Moreover, this study provides support for the notion that there is a greater influence of perceived value on behavioral intention compared to satisfaction [85]. Specifically, visitors perceive online digital art exhibitions as possessing high literary and historical value, making them worth visiting. Visitors' behavioral intentions are more strongly driven by perceived value than satisfaction. Additionally, the inclusion of art therapy as a predictor of visitors' on-site visitation behavior represents a novel discovery in this study. Previous research suggests that experiencing a healing sensation from a favorable environmental stimulus strongly affects visitors' attachment to a place and behavioral intentions [86]. This indicates that the confirmation of visitors' emotions arising from the art therapy effect of online cultural heritage digital art exhibitions can effectively predict their intention to engage in on-site visitation and their attachment to the art exhibition. Although COVID-19 is gradually receding, the psychological trauma caused by the pandemic lingers and cannot be easily alleviated. However, through immersive online experiences that allow visitors to immerse themselves in Song and Yuan aesthetics, appreciate the daily culture of the Song Dynasty spanning thousands of years, interact with the exhibits, and learn new information, individuals can effectively address emotions such as anxiety, depression, and anger, thereby promoting psychological healing. Consequently, visitors are more inclined to visit offline museums and personally experience the spiritual healing that art offers.

## 6. Conclusions and Suggestions

Firstly, this study establishes a new S-O-R model based on the Expectation Confirmation Model, perceived quality, and art therapy theory. The model explains 48% of the variance in visitors' behavioral intentions to engage in on-site visitation, providing a cognitive framework to analyze the logic underlying the interaction between the perceived quality of online digital art exhibitions and on-site visitation behavior. It delves into the stimuli of perceived quality experienced by visitors through online digital art exhibitions, leading to perceptions of increased value, enjoyment, and art therapy, thereby influencing visitors' decisions to engage in physical visits. Therefore, the model developed in this study serves as empirical evidence of the benefits of using digital technology in tourism experiences and the communication of cultural heritage tourism.

Secondly, we offer an effective method to assess whether online digital art exhibitions meet visitors' expectations regarding enjoyment, value, and art therapy, thereby enhancing their likelihood of engaging in on-site visitation behavior. These research findings will enable cultural heritage tourism managers to understand visitors' motivations, allowing them to design online digital art exhibitions that cater to visitors' psychological needs and encourage them to visit on-site. This is particularly significant in the aftermath of the COVID-19 pandemic. In fact, online digital art exhibitions provide an alternative for individuals who may be interested in art exhibition content but are unable to physically attend. Furthermore, our research results may encourage web designers of online digital art exhibitions to improve the usability and aesthetics of their online interfaces. By effectively utilizing the Internet, multimedia, emotional content, and aesthetic interfaces, online digital art exhibitions can attract a larger audience and generate interest in visiting art exhibitions and environments within the cultural domain. In the digital information age, online cultural heritage art exhibitions bear the social mission of informing the public about historical civilizations and preserving cultural heritage.

Firstly, this study could provide practical insights for tourism managers, as online digital cultural tourism is a key driver in the transformation of the culture and tourism industry. With the accelerated process of digitization, blockchain-based digital art has become a new trend in the wave of digital economic development [87]. The "digital + culture" development model presents infinite possibilities for the revitalization of China's remarkable culture and the development of cultural tourism. Digital transformation has promoted the transformation and upgrading of the tourism industry. Online digital art ex-

hibitions, representing a transition from traditional offline exhibition formats to intelligent online travel experiences, are propelling the tourism industry towards greater efficiency and intelligence.

Secondly, this study provides guidance for tourism managers on the bidirectional integration of online and offline exhibition technologies. Online exhibitions focus on personalized services, offering visitors the opportunity to access and enjoy cultural and artistic activities anytime, while reducing the use of travel resources and associated environmental pressures such as carbon emissions from air travel. It significantly reduces the energy costs associated with maintaining buildings, hosting exhibitions, and travel. This is particularly beneficial for those who have limited mobility or lack economic resources for travel, thereby promoting the sustainability of the tourism industry. On the other hand, offline exhibitions emphasize quality and provide unique tourism value through interactive settings and real-time experiences. Visitor engagement in cultural and artistic activities can stimulate the local community's economic and cultural life. The integration of online and offline experiences mutually reinforces the sustainability of the tourism industry.

Thirdly, this study contributes to the understanding of how online digital technologies can revitalize historical and cultural heritage in the field of culture and tourism management. Online digital exhibitions allow developers to vividly present information on historical and cultural heritage and folk culture, urban stories, and brand concepts to visitors. This not only promotes the vitality of traditional culture and tourism industry values and commercial values, but it also satisfies the public's curiosity and thirst for knowledge through interactive experiences using digital technology. Thus, cultural heritage is effectively protected while being showcased and utilized, providing more inspiration and forms of expression for cultural heritage inheritance and development.

Furthermore, this research contributes to enhancing citizens' well-being. In the context of the post-pandemic era, online digital art exhibitions can serve as healing spaces, alleviating the pressures of temporary living, working, and studying, and nurturing psychological well-being. In this sense, online digital art exhibitions deserve attention from curators and designers. Unrestricted by space and distance, online digital art exhibitions have a large audience and wide dissemination, making them the fastest and most effective platforms for art therapy. Today's online digital art exhibitions extend beyond simply replicating offline museum and art gallery experiences. Through the integration of technology and art, they emphasize individual spiritual pleasure and freedom. Cognitive psychology also demonstrates that high-quality interactive art is more likely to enable viewers to contemplate their own existence, experience basic sensory and behavioral responses, and awaken deep emotional resonance [88]. By guiding audiences from the real world to the virtual world, a dialogue between individuals and artworks is accomplished, incorporating art exhibitions into users' lifestyles and elevating the overall spiritual and cultural development of society.

This study has several limitations that should be acknowledged. Firstly, digital technology tourism is developing rapidly worldwide, but research on this topic in China is still in its early stages. Additionally, the sample used in this study may lack diversity across ethnic and cultural backgrounds, as visitor behavior may be influenced by their ethnic culture. Therefore, the generalizability of our research findings needs to be extended to other countries with well-developed online digital exhibitions and different ethnic cultures. Secondly, due to time constraints, we relied on cross-sectional experiential data to measure customer perceptions and behavior. The internal changes that occurred among visitors remain unknown. Considering the ongoing evolution of online digital exhibition content, features, and services, as well as the advancements in technology, measurement biases resulting from time gaps may be increased. Hence, we recommend that future researchers, given sufficient funding and time, collect longitudinal empirical data to examine the interactions among different temporal variables, thereby obtaining more effective and robust validation results.

**Funding:** This research received no external funding.



**Institutional Review Board Statement:** Not applicable.

**Informed Consent Statement:** Not applicable.

**Data Availability Statement:** Not applicable.

**Acknowledgments:** The authors would like to express their sincere gratitude to the reviewers and Editor for their valuable suggestions, and the College of Art and Design of Zhejiang Guangsha Vocational and Technical University of construction and Pukyong National University for their strong support of this research.

**Conflicts of Interest:** The authors declare no conflict of interest.

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
