# Peer review of "How Has Online Digital Technology Influenced the On-Site Visitation Behavior of Tourists during the COVID-19 Pandemic? A Case Study of Online Digital Art Exhibitions in China"

_sustainability, doi:10.3390/su151410889_

Round 1
Reviewer 1 Report
Dear authors,
It was really a pleasure to read the article in its current state.
The only criticism I can make is related to the title itself. You can clearly do better and allow your article a more concrete positioning.
Good luck!
Author Response
Point 1:
Dear authors,
It was really a pleasure to read the article in its current state.
The only criticism I can make is related to the title itself. You can clearly do better and allow your article a more concrete positioning.Good luck!
1.Response to reviewer:Your recognition of our research is much appreciated. After carefully reviewing your comments, the authors of this study have deliberated and reflected on the paper's content and title. Indeed, modifying the title will better reflect the research findings. We sincerely appreciate your valuable suggestions.
The modified title of this article is as follows:
How has online digital technology influenced the on-site visitation behavior of tourists during the COVID-19 pandemic? A case study of online digital art exhibitions in China
- Response to Reviewer:Thanks for your recognition and sincere comments. With your help, we have a more quality paper, and your comments inspired us greatly. Again, thanks for your kindness.

Reviewer 2 Report
Online exhibitions can have a significant impact on the sustainability of tourism. The paper sought to draw conclusions on the relationship between post-pandemic tourism recovery and sustainability from the virtual experience of online exhibitions, the link between perceived enjoyment and art therapy, and a questionnaire survey of visitor satisfaction surveys.
The research methodology, sampling, hypotheses were clearly presented. The evaluation results of the questionnaire are acceptable. However, conclusions on sustainable tourism planning are hardly presented.
One of the advantages of online exhibitions is that they facilitate virtual visits and participation in exhibitions wherever one is in the world. This can reduce the need to travel and associated environmental pressures such as emissions from air travel. Online exhibitions offer tourists the opportunity to learn about and enjoy cultural and artistic events without having to attend in person. This can be particularly beneficial for those with limited mobility or who do not have the financial resources to travel.
Online exhibitions can contribute to the sustainability of tourism by reducing resource use and environmental impact. The energy costs of maintaining buildings, setting up exhibitions and travelling can be significantly reduced. In addition, online platforms offer the possibility to digitise content, which can contribute to the preservation and accessibility of cultural heritage.
However, it is important to note that online exhibitions cannot fully replace the traditional, in-person experience. Personal interaction, real-time experiences and sensory experiences can provide unique value for tourists and visitors. Personal participation in cultural and artistic events can contribute to the economy and cultural life of local communities.
I suggest that the proposals for sustainability are set out in detail in the conclusions.
Author Response
Point 1:
I suggest that the proposals for sustainability are set out in detail in the conclusions.
Online exhibitions can have a significant impact on the sustainability of tourism. The paper sought to draw conclusions on the relationship between post-pandemic tourism recovery and sustainability from the virtual experience of online exhibitions, the link between perceived enjoyment and art therapy, and a questionnaire survey of visitor satisfaction surveys.
The research methodology, sampling, hypotheses were clearly presented. The evaluation results of the questionnaire are acceptable. However, conclusions on sustainable tourism planning are hardly presented.
One of the advantages of online exhibitions is that they facilitate virtual visits and participation in exhibitions wherever one is in the world. This can reduce the need to travel and associated environmental pressures such as emissions from air travel. Online exhibitions offer tourists the opportunity to learn about and enjoy cultural and artistic events without having to attend in person. This can be particularly beneficial for those with limited mobility or who do not have the financial resources to travel.
Online exhibitions can contribute to the sustainability of tourism by reducing resource use and environmental impact. The energy costs of maintaining buildings, setting up exhibitions and travelling can be significantly reduced. In addition, online platforms offer the possibility to digitise content, which can contribute to the preservation and accessibility of cultural heritage.
However, it is important to note that online exhibitions cannot fully replace the traditional, in-person experience. Personal interaction, real-time experiences and sensory experiences can provide unique value for tourists and visitors. Personal participation in cultural and artistic events can contribute to the economy and cultural life of local communities.
Response to Reviewer:Your recognition of our research is much appreciated. Thank you for the constructive suggestion, which we found very helpful.As a result, we have added a new chapter on research implications and recommendations. Additionally, in Section 6.2, we have included content related to sustainable tourism planning, which has significantly enhanced the quality of the article. Once again, we sincerely appreciate your insightful suggestions.
The following is a list of suggestions for improving the paper:
6.Conclusions and Suggestions
6.1.Theoretical implications
Firstly, this study establishes a new S-O-R model based on the expectation confirmation model, perceived quality, and art therapy theory. The model explains 48% of the variance in visitors' behavioral intentions to engage in on-site visitation, providing a cognitive framework to analyze the logic underlying the interaction between the perceived quality of online digital art exhibitions and on-site visitation behavior. It delves into the stimuli of perceived quality experienced by visitors through online digital art exhibitions, leading to perceptions of increased value, enjoyment, and art therapy, thereby influencing visitors' decisions to engage in physical visits. Therefore, the model developed in this study serves as empirical evidence of the benefits of using digital technology in tourism experiences and the communication of cultural heritage tourism.
Secondly, we offer an effective method to assess whether online digital art exhibitions meet visitors' expectations regarding enjoyment, value, and art therapy, thereby enhancing their likelihood of engaging in on-site visitation behavior. These research findings will enable cultural heritage tourism managers to understand visitors' motivations, allowing them to design online digital art exhibitions that cater to visitors' psychological needs and encourage them to visit on-site. This is particularly significant in the aftermath of the COVID-19 pandemic. In fact, online digital art exhibitions provide an alternative for individuals who may be interested in art exhibition content but are unable to physically attend. Furthermore, our research results may encourage web designers of online digital art exhibitions to improve the usability and aesthetics of their online interfaces. By effectively utilizing the Internet, multimedia, emotional content, and aesthetic interfaces, online digital art exhibitions can attract a larger audience and generate interest in art exhibitions and visiting environments within the cultural domain. In the digital information age, online cultural heritage art exhibitions bear the social mission of informing the public about historical civilizations and preserving cultural heritage.
6.2. Practical Inspiration
Firstly, this study could provide practical insights for tourism managers, as online digital cultural tourism is a key driver in the transformation of the culture and tourism industry. With the accelerated process of digitization, blockchain-based digital art has become a new trend in the wave of digital economic development [87]. The "digital + culture" development model presents infinite possibilities for the revitalization of China's remarkable culture and the development of cultural tourism. Digital transformation has promoted the transformation and upgrading of the tourism industry. Online digital art exhibitions, representing a transition from traditional offline exhibition formats to intelligent online travel experiences, are propelling the tourism industry towards greater efficiency and intelligence.
Secondly, this study provides guidance for tourism managers on the bidirectional integration of online and offline exhibition technologies. Online exhibitions focus on personalized services, offering visitors the opportunity to access and enjoy cultural and artistic activities anytime, while reducing the use of travel resources and associated environmental pressures such as carbon emissions from air travel. It significantly reduces the energy costs associated with maintaining buildings, hosting exhibitions, and travel. This is particularly beneficial for those who have limited mobility or lack economic resources for travel, thereby promoting the sustainability of the tourism industry. On the other hand, offline exhibitions emphasize quality and provide unique tourism value through interactive settings and real-time experiences. Visitor engagement in cultural and artistic activities can stimulate the local community's economic and cultural life. The integration of online and offline experiences mutually reinforces the sustainability of the tourism industry.
Thirdly, this study contributes to the understanding of how online digital technologies can revitalize historical and cultural heritage in the field of culture and tourism management. Online digital exhibitions allow developers to vividly present information on historical and cultural heritage and folk culture, urban stories, and brand concepts to visitors. This not only promotes the vitality of traditional culture and tourism industry values and commercial values, but also satisfies the public's curiosity and thirst for knowledge through interactive experiences using digital technology. Thus, cultural heritage is effectively protected while being showcased and utilized, providing more inspiration and forms of expression for cultural heritage inheritance and development.
Furthermore, this research contributes to enhancing citizens' well-being. In the context of the post-pandemic era, online digital art exhibitions can serve as healing spaces, alleviating the pressures of temporary living, working, and studying, and nurturing psychological well-being. In this sense, online digital art exhibitions deserve attention from curators and designers. Unrestricted by space and distance, online digital art exhibitions have a large audience and wide dissemination, making them the fastest and most effective platforms for art therapy. Today's online digital art exhibitions extend beyond simply replicating offline museum and art gallery experiences. Through the integration of technology and art, they emphasize individual spiritual pleasure and freedom. Cognitive psychology also demonstrates that high-quality interactive art is more likely to enable viewers to contemplate their own existence, experience basic sensory and behavioral responses, and awaken deep emotional resonance [88]. By guiding audiences from the real world to the virtual world, a dialogue between individuals and artworks is accomplished, incorporating art exhibitions into users' lifestyles and elevating the overall spiritual and cultural development of society.
2.In response to the language issues mentioned in your review report, we have promptly engaged professional experts to revise the paper.

Response to Reviewer:Thanks for your recognition and sincere comments. With your help, we have a more quality paper, and your comments inspired us greatly. Again, thanks for your kindness.

Reviewer 3 Report
In order to be considered for publication in any form, the article needs to be thoroughly re-written.
Firstly, the title of the article and the article content do not completely address each other. The title announces the discussion on the changes in tourist industry after the COVID-19, while the article discusses the pull factors which would attract online visitors to visit ‘live’ art, after being there online. The two are related, of course, but the focus on the article is on the ways and means the museums could attract more visitors in the future, due to financing, different experience, and not on the force majeure conditions which forced the art institutions to act differently.
Even though the article topic is relevant and worth researching, the article however goes on to discuss and analyze two rather separate points: 1) the experience of online art exhibitions/ museum visits of virtual tourists and 2) art therapy, which is a complex subject in itself. In order for the article to be coherent, the author should choose one or the other. I suggest, the first one, since the second one, art therapy, is not sufficiently tackled.
Furthermore, the article is difficult to read, somewhat with repetitive segments and with some arguments repeated throughout the paragraphs, which is rather confusing. Thus, even though present, the logic of the arguments gets lost in the repetitions and wordiness. Also, some of the conclusions seem a little bit too simple, like the one that the museums should pay attention to web-page and applications design to attract visitors, since this topic has been ‘around’ in the museum studies for decades now.
In conclusion, the topic is relevant and the article could be re-written.
Extensive editing is required.
Author Response
Point 1:
In order to be considered for publication in any form, the article needs to be thoroughly re-written.
Response to Reviewer:Thank you very much for your valuable comments on this paper. Your input has been immensely helpful to us. The authors of this study have thoroughly reviewed your research and, based on your suggestions, our research team engaged in in-depth discussions and made extensive modifications to the title, abstract, introduction, conclusion, and discussion sections of the article. The specific modifications are as follows:
- Firstly, the title of the article and the article content do not completely address each other. The title announces the discussion on the changes in tourist industry after the COVID-19, while the article discusses the pull factors which would attract online visitors to visit ‘live’ art, after being there online. The two are related, of course, but the focus on the article is on the ways and means the museums could attract more visitors in the future, due to financing, different experience, and not on the force majeure conditions which forced the art institutions to act differently.
Response to Reviewer:Thank you for your sincere suggestions. The research team has carefully considered the title and content of the paper and indeed identified several areas that required improvement. As a result, we have made profound modifications to both the title and content of the article, aiming to clarify the research problem and research approach. The specific modifications are as follows:
How has online digital technology influenced the on-site visitation behavior of tourists during the COVID-19 pandemic? A case study of online digital art exhibitions in China
Abstract: The COVID-19 pandemic has had a significant impact on the global tourism industry, leading to a decrease in peoples’ willingness to travel and a sense of insecurity regarding tourist destinations. Therefore, restoring people’s willingness to travel is the greatest challenge faced by this industry in the post-pandemic era. The tourism industry requires innovative solutions to achieve sustainable recovery. While there is a considerable amount of research on its recovery during the pandemic, there are few studies exploring people’s willingness to travel to encourage sustainable and resilient recovery in the post-pandemic era. This study employed a quality model to examine the satisfaction and intention of tourists towards the application of online digital art exhibitions under the influence of COVID-19. The aim was to investigate the promoting role of online digital art exhibitions in the sustainability and resilient recovery of the tourism industry. To achieve these objectives, this study focuses on the online digital art exhibition of Song Dynasty figure paintings launched by China Central Television (CCTV), with post-exhibition surveys conducted and 512 valid questionnaires collected. The research model and hypotheses are tested using structural equation modeling. The results of this study indicate that travelers' intentions to engage in on-site visits through online digital exhibitions are determined by three factors: perceived value, satisfaction, and art therapy. Furthermore, online digital art exhibitions not only represent the most important form of tourism during the pandemic, but also provide significant psychological healing. They have become a driving force for the transformation of the current culture and tourism industry and the promotion of its sustainable development. This research provides a benchmark for future research on the tourism industry, and offers new research directions in the field of sustainable tourism.
Keywords: sustainable tourism; tourism industry; COVID-19; behavioral intention; online digital exhibition
- Introduction
The tourism industry provides livelihoods to millions of people and enables billions to appreciate diverse cultures and the natural world. In some countries, tourism accounts for more than 20% of the gross domestic product (GDP), making it the third-largest export sector in the global economy [1]. However, the COVID-19 pandemic has severely impacted the tourism industry, affecting economies, livelihoods, and public services across continents and bringing the industry to a temporary halt [2]. According to forecasts by the World Travel & Tourism Council (WTTC), the COVID-19 pandemic was expected to cause a loss in global tourism value of USD 2.1 trillion in 2020 and jeopardize 75 million tourism jobs [3]. Rebuilding market confidence, revitalizing the tourism economy, and achieving high-quality development of the tourism industry have become focal points in academia and industry post-pandemic.
The outbreak of the pandemic has posed unprecedented challenges to the tourism industry, yet it has also presented an opportunity for transformation and change. The emergence of online digital technologies has facilitated the digital transformation and innovation of the tourism sector.[4]. Digital cultural tourism initiatives have emerged during the pandemic, leveraging new-generation technologies such as virtual reality (VR), augmented reality (AR), and artificial intelligence (AI) to create immersive experiences. These initiatives include virtual reality-based scenic spots, entertainment, and museums, and other new forms of culture and tourism experience, fostering new consumer behavior and reshaping people's travel preferences and tourism concepts [5]. Emphasizing safety in tourism and the ability to explore the world from home, and prioritizing user experience, have become paramount [5]. Under the circumstances of the pandemic, online digital technologies have gained more attention than ever from tourists and destination organizations. Online virtual tours of museums and exhibitions and other forms of "cloud tourism" have become the new norm since the pandemic, stimulating people's interest in tourism. Concurrently, online digital technologies are reshaping the manner in which customers plan their travel and search for destination information. By leveraging digitized environments, these technologies enable customers to experience products, services, or locations prior to their physical visit[6]. The increasing reliance on and desire for such online digital environments reflects a gradual shift in tourism perspectives, with individuals predominantly opting for online virtual tourism. Consequently, online digital technologies have opened up new avenues for the revitalization and recovery of the tourism industry, providing innovative approaches to travel and tourism experiences.
Previous studies have explored the use of on-site digital technologies [7] and VR technology for enhancing tourists' travel experiences [8], as well as the impact of VR technology on intentions to make repeat visits to tourist destinations [9-10]. However, the perspectives explored in these studies have been relatively narrow, focusing solely on the impact of digital technologies on tourism destinations. They predominantly concentrate on the "technology" itself, without delving into the influence of online digital technologies on "people" after the pandemic. Particularly, the shift in individuals' tourism perspectives and their heightened concerns for personal well-being and health have significantly dampened their willingness to travel. Yet, the revival of tourist demand is considered a key factor in stimulating the recovery of the tourism industry[11] . Consequently, there is an urgent need to address how to effectively guide tourists in regaining their willingness to travel. In fact, tourists are highly interested in online digital technologies and seek to understand how these platforms can provide humanistic care[12-13]. However, there have been few attempts to evaluate user satisfaction with online digital technology applications and their subsequent impact on travel intentions. How do online digital technologies influence tourists' on-site visitation behavior during the COVID-19 pandemic? What are the underlying factors affecting this behavior? Do they contribute to stimulating tourists' travel intentions and promoting the recovery and sustainable development of the tourism industry in the post-pandemic era?
To address the aforementioned issues, this study introduces a comprehensive research model that incorporates factors derived from perceived quality (content, system, and personalized service) and expectation confirmation. From the perspective of tourists' expectation confirmation, the research model explores the effects of perceived quality of online digital art exhibitions, user expectation confirmation, perceived value, perceived enjoyment, and art therapy on satisfaction with online digital art exhibitions and on-site visitation. By examining how online digital technology influences the physical visit behavior of tourists during the COVID-19 pandemic, this research can assist museum curators and other cultural tourism institutions in leveraging digital technology to increase public engagement with art, promote the digital transformation of the culture and tourism industry, stimulate innovation in this sector, unleash the multiplicative effects of digitization on the industry, and facilitate its sustainable development.
Point2: Even though the article topic is relevant and worth researching, the article however goes on to discuss and analyze two rather separate points: 1) the experience of online art exhibitions/ museum visits of virtual tourists and 2) art therapy, which is a complex subject in itself. In order for the article to be coherent, the author should choose one or the other. I suggest, the first one, since the second one, art therapy, is not sufficiently tackled.
Response to Reviewer:We greatly appreciate your sincere suggestions. Indeed, this research should focus on a central issue for in-depth analysis. Based on the empirical findings of our study, we have specifically added a discussion section to explore how the future online art exhibition experience can attract visitors, contribute to the development of the tourism industry, and provide relevant strategies for its growth. This, in turn, aims to promote sustainable development in the tourism industry. The specific modifications are as follows:
6.2. Practical Inspiration
Firstly, this study could provide practical insights for tourism managers, as online digital cultural tourism is a key driver in the transformation of the culture and tourism industry. With the accelerated process of digitization, blockchain-based digital art has become a new trend in the wave of digital economic development [87]. The "digital + culture" development model presents infinite possibilities for the revitalization of China's remarkable culture and the development of cultural tourism. Digital transformation has promoted the transformation and upgrading of the tourism industry. Online digital art exhibitions, representing a transition from traditional offline exhibition formats to intelligent online travel experiences, are propelling the tourism industry towards greater efficiency and intelligence.
Secondly, this study provides guidance for tourism managers on the bidirectional integration of online and offline exhibition technologies. Online exhibitions focus on personalized services, offering visitors the opportunity to access and enjoy cultural and artistic activities anytime, while reducing the use of travel resources and associated environmental pressures such as carbon emissions from air travel. It significantly reduces the energy costs associated with maintaining buildings, hosting exhibitions, and travel. This is particularly beneficial for those who have limited mobility or lack economic resources for travel, thereby promoting the sustainability of the tourism industry. On the other hand, offline exhibitions emphasize quality and provide unique tourism value through interactive settings and real-time experiences. Visitor engagement in cultural and artistic activities can stimulate the local community's economic and cultural life. The integration of online and offline experiences mutually reinforces the sustainability of the tourism industry.
Thirdly, this study contributes to the understanding of how online digital technologies can revitalize historical and cultural heritage in the field of culture and tourism management. Online digital exhibitions allow developers to vividly present information on historical and cultural heritage and folk culture, urban stories, and brand concepts to visitors. This not only promotes the vitality of traditional culture and tourism industry values and commercial values, but also satisfies the public's curiosity and thirst for knowledge through interactive experiences using digital technology. Thus, cultural heritage is effectively protected while being showcased and utilized, providing more inspiration and forms of expression for cultural heritage inheritance and development.
Point 3: Furthermore, the article is difficult to read, somewhat with repetitive segments and with some arguments repeated throughout the paragraphs, which is rather confusing. Thus, even though present, the logic of the arguments gets lost in the repetitions and wordiness. Also, some of the conclusions seem a little bit too simple, like the one that the museums should pay attention to web-page and applications design to attract visitors, since this topic has been ‘around’ in the museum studies for decades now.
Response to Reviewer:We sincerely appreciate your valuable suggestions. In response, we have made the following modifications.In Chapter 2, we have eliminated redundant and repetitive content, extracting complex theories from the text to focus on the main theme and structure of the article. We have clarified the logical flow of the paper and added a conclusion section in Chapter 6 to emphasize the theoretical and practical significance, enriching the depth of the conclusions.
6.Conclusions and Suggestions
6.1.Theoretical implications
Firstly, this study establishes a new S-O-R model based on the expectation confirmation model, perceived quality, and art therapy theory. The model explains 48% of the variance in visitors' behavioral intentions to engage in on-site visitation, providing a cognitive framework to analyze the logic underlying the interaction between the perceived quality of online digital art exhibitions and on-site visitation behavior. It delves into the stimuli of perceived quality experienced by visitors through online digital art exhibitions, leading to perceptions of increased value, enjoyment, and art therapy, thereby influencing visitors' decisions to engage in physical visits. Therefore, the model developed in this study serves as empirical evidence of the benefits of using digital technology in tourism experiences and the communication of cultural heritage tourism.
Secondly, we offer an effective method to assess whether online digital art exhibitions meet visitors' expectations regarding enjoyment, value, and art therapy, thereby enhancing their likelihood of engaging in on-site visitation behavior. These research findings will enable cultural heritage tourism managers to understand visitors' motivations, allowing them to design online digital art exhibitions that cater to visitors' psychological needs and encourage them to visit on-site. This is particularly significant in the aftermath of the COVID-19 pandemic. In fact, online digital art exhibitions provide an alternative for individuals who may be interested in art exhibition content but are unable to physically attend. Furthermore, our research results may encourage web designers of online digital art exhibitions to improve the usability and aesthetics of their online interfaces. By effectively utilizing the Internet, multimedia, emotional content, and aesthetic interfaces, online digital art exhibitions can attract a larger audience and generate interest in art exhibitions and visiting environments within the cultural domain. In the digital information age, online cultural heritage art exhibitions bear the social mission of informing the public about historical civilizations and preserving cultural heritage.
6.2. Practical Inspiration
Firstly, this study could provide practical insights for tourism managers, as online digital cultural tourism is a key driver in the transformation of the culture and tourism industry. With the accelerated process of digitization, blockchain-based digital art has become a new trend in the wave of digital economic development [87]. The "digital + culture" development model presents infinite possibilities for the revitalization of China's remarkable culture and the development of cultural tourism. Digital transformation has promoted the transformation and upgrading of the tourism industry. Online digital art exhibitions, representing a transition from traditional offline exhibition formats to intelligent online travel experiences, are propelling the tourism industry towards greater efficiency and intelligence.
Secondly, this study provides guidance for tourism managers on the bidirectional integration of online and offline exhibition technologies. Online exhibitions focus on personalized services, offering visitors the opportunity to access and enjoy cultural and artistic activities anytime, while reducing the use of travel resources and associated environmental pressures such as carbon emissions from air travel. It significantly reduces the energy costs associated with maintaining buildings, hosting exhibitions, and travel. This is particularly beneficial for those who have limited mobility or lack economic resources for travel, thereby promoting the sustainability of the tourism industry. On the other hand, offline exhibitions emphasize quality and provide unique tourism value through interactive settings and real-time experiences. Visitor engagement in cultural and artistic activities can stimulate the local community's economic and cultural life. The integration of online and offline experiences mutually reinforces the sustainability of the tourism industry.
Thirdly, this study contributes to the understanding of how online digital technologies can revitalize historical and cultural heritage in the field of culture and tourism management. Online digital exhibitions allow developers to vividly present information on historical and cultural heritage and folk culture, urban stories, and brand concepts to visitors. This not only promotes the vitality of traditional culture and tourism industry values and commercial values, but also satisfies the public's curiosity and thirst for knowledge through interactive experiences using digital technology. Thus, cultural heritage is effectively protected while being showcased and utilized, providing more inspiration and forms of expression for cultural heritage inheritance and development.
Furthermore, this research contributes to enhancing citizens' well-being. In the context of the post-pandemic era, online digital art exhibitions can serve as healing spaces, alleviating the pressures of temporary living, working, and studying, and nurturing psychological well-being. In this sense, online digital art exhibitions deserve attention from curators and designers. Unrestricted by space and distance, online digital art exhibitions have a large audience and wide dissemination, making them the fastest and most effective platforms for art therapy. Today's online digital art exhibitions extend beyond simply replicating offline museum and art gallery experiences. Through the integration of technology and art, they emphasize individual spiritual pleasure and freedom. Cognitive psychology also demonstrates that high-quality interactive art is more likely to enable viewers to contemplate their own existence, experience basic sensory and behavioral responses, and awaken deep emotional resonance [88]. By guiding audiences from the real world to the virtual world, a dialogue between individuals and artworks is accomplished, incorporating art exhibitions into users' lifestyles and elevating the overall spiritual and cultural development of society.
6.3. Limitations and Future Work
This study has several limitations that should be acknowledged. Firstly, digital technology tourism is developing rapidly worldwide, but research on this topic in China is still in its early stages. Additionally, the sample used in this study may lack diversity across ethnic and cultural backgrounds, as visitor behavior may be influenced by their ethnic culture. Therefore, the generalizability of our research findings needs to be extended to other countries with well-developed online digital exhibitions and different ethnic cultures. Secondly, due to time constraints, we relied on cross-sectional experiential data to measure customer perceptions and behavior. The internal changes that occurred among visitors remain unknown. Considering the ongoing evolution of online digital exhibition content, features, and services, as well as the advancements in technology, measurement biases resulting from time gaps may be increased. Hence, we recommend that future researchers, given sufficient funding and time, collect longitudinal empirical data to examine the interactions among different temporal variables, thereby obtaining more effective and robust validation results.

Response to Reviewer:Furthermore, to avoid instances where the language in the paper may lead to unclear or ambiguous points, we have engaged professional English language experts to revise the paper after the modifications. This step ensures the accuracy of the language used in the paper.
With best regards
Response to Reviewer:Thanks for your recognition and sincere comments. With your help, we have a more quality paper, and your comments inspired us greatly. Again, thanks for your kindness.

Reviewer 4 Report
The manuscript should not have author identification. There are several situations in the formal plan of the template that should be improved for later publication (e.g. line 211, 241, 270, 284, 694, etc.). On line 694 it seems that the text is either incomplete or has been inadvertently cut.
Author Response
Point 1:
The manuscript should not have author identification.
Response to reviewer:Your recognition of our research is much appreciated. After carefully reviewing your feedback, the authors of this study have thoroughly examined the publication requirements of the journal and made necessary modifications. We sincerely appreciate your valuable suggestions.
2.There are several situations in the formal plan of the template that should be improved for later publication (e.g. line 211, 241, 270, 284, 694, etc.). On line 694 it seems that the text is either incomplete or has been inadvertently cut.
Response to reviewer:Thank you for your sincere comment. The members of our research team have once again verified whether the journal's revision requirements were followed and addressed each of the raised issues individually. In particular, we have undertaken a reorganization and revision of the content, specifically addressing the incoherent and seemingly irrelevant sections as mentioned in the previous version of the paper (694). Additionally, we have engaged professional English language editors to modify the language of the paper, ensuring that there are no longer any incomplete or ambiguous aspects in the content.The specific modifications are as follows:
6.2. Practical Inspiration
Firstly, this study could provide practical insights for tourism managers, as online digital cultural tourism is a key driver in the transformation of the culture and tourism industry. With the accelerated process of digitization, blockchain-based digital art has become a new trend in the wave of digital economic development [87]. The "digital + culture" development model presents infinite possibilities for the revitalization of China's remarkable culture and the development of cultural tourism. Digital transformation has promoted the transformation and upgrading of the tourism industry. Online digital art exhibitions, representing a transition from traditional offline exhibition formats to intelligent online travel experiences, are propelling the tourism industry towards greater efficiency and intelligence.
Secondly, this study provides guidance for tourism managers on the bidirectional integration of online and offline exhibition technologies. Online exhibitions focus on personalized services, offering visitors the opportunity to access and enjoy cultural and artistic activities anytime, while reducing the use of travel resources and associated environmental pressures such as carbon emissions from air travel. It significantly reduces the energy costs associated with maintaining buildings, hosting exhibitions, and travel. This is particularly beneficial for those who have limited mobility or lack economic resources for travel, thereby promoting the sustainability of the tourism industry. On the other hand, offline exhibitions emphasize quality and provide unique tourism value through interactive settings and real-time experiences. Visitor engagement in cultural and artistic activities can stimulate the local community's economic and cultural life. The integration of online and offline experiences mutually reinforces the sustainability of the tourism industry.
Thirdly, this study contributes to the understanding of how online digital technologies can revitalize historical and cultural heritage in the field of culture and tourism management. Online digital exhibitions allow developers to vividly present information on historical and cultural heritage and folk culture, urban stories, and brand concepts to visitors. This not only promotes the vitality of traditional culture and tourism industry values and commercial values, but also satisfies the public's curiosity and thirst for knowledge through interactive experiences using digital technology. Thus, cultural heritage is effectively protected while being showcased and utilized, providing more inspiration and forms of expression for cultural heritage inheritance and development.
Furthermore, this research contributes to enhancing citizens' well-being. In the context of the post-pandemic era, online digital art exhibitions can serve as healing spaces, alleviating the pressures of temporary living, working, and studying, and nurturing psychological well-being. In this sense, online digital art exhibitions deserve attention from curators and designers. Unrestricted by space and distance, online digital art exhibitions have a large audience and wide dissemination, making them the fastest and most effective platforms for art therapy. Today's online digital art exhibitions extend beyond simply replicating offline museum and art gallery experiences. Through the integration of technology and art, they emphasize individual spiritual pleasure and freedom. Cognitive psychology also demonstrates that high-quality interactive art is more likely to enable viewers to contemplate their own existence, experience basic sensory and behavioral responses, and awaken deep emotional resonance [88]. By guiding audiences from the real world to the virtual world, a dialogue between individuals and artworks is accomplished, incorporating art exhibitions into users' lifestyles and elevating the overall spiritual and cultural development of society.
- The manuscript has been revised by English professionals

With best regards
Response to Reviewer:Thanks for your recognition and sincere comments. With your help, we have a more quality paper, and your comments inspired us greatly. Again, thanks for your kindness.

Reviewer 5 Report
We unfortunately feel inclined to wonder about the appropriateness of this article’s inclusion (in the article’s current state) in the scholarly realm. While the author(s) probably rightly note the understudied nature of the so-called integration of online and offline platforms in the abstract, this paper serves as a reminder of the need for ingenuity and uniqueness during the discernment process that accompanies a researcher’s formulation of a research problem. Sometimes, a researcher decides to embark on a project that has all the appearances of appearing obviously or self-evidently inconsequential; even an infinite number of second glances would not transform those projects into matters of greater importance. Some researchers feel that these obviously inconsequential projects have never had any previous investigators, so such projects must naturally define viable research topics. The takeaway of this discussion lies in the fact that the mere novelty of a project does not turn that topic into a viable research project. The introduction also left this reviewer utterly confused as to what the researcher(s) wanted to demonstrate or prove, and this confusion probably had something to do with the opacity of the prose and the fact that the introduction seemed extremely discursive (i.e., jumping around so many topics in lengthy paragraphs whose topic sentences gave little clue over the content of the paragraphs).
In order to avoid the abovementioned issue(s), the author(s) should strongly consider a paper of an altogether different topic, namely an exploration of the nuances of how cultural heritage tourism administrators actually structure online exhibitions. In other words, one would want to know the reasons that animate individual cultural heritage tourism administrators to make certain selections regarding the intricacies of online exhibitions. We already know the basic reasons for the implementation of online exhibitions (i.e., a need to stay financially stable and to make cultural artifacts available for pandemic-restricted audiences, and so forth), but we wonder over the reasons that animate exhibitors to digitize one exhibition over another exhibition (i.e., a Song Dynasty-related piece, as the authors explain, over, say, a Tang-Dynasty piece). By delving into the mindsets and attitudes of museum administrators, the author(s) can write a far more illuminating paper devoid of the overly and unnecessarily abstruse theoretical jargon that impregnates a draft like this one---a draft that seems more like a sterile compilation of data than an actual attempt to analyze the data in interdisciplinary lenses that range from history to ethnography to cultural anthropology to other academic disciplines represented by museums. Such a paper would have a solid foundation in the disciplines of philosophy, history, anthropology, and archaeology---and the data of such a piece would serve as a handmaiden of the interpretive lenses of the author(s). As it now stands, the paper gives off an impression of wanting to simply stand on the data and little else. The current paper would probably have a more clearly defined and necessary role in the research and development committee of a government think tank, but not necessarily a scholarly journal.
For some miscellaneous remarks, we should like to add the following. The paper oddly ends at line 694 in a sentence that does not terminate with a punctuation mark, suggesting that the authors wanted to go on with some further discussion that never quite materialized. In addition, the paper indicates a single author, but the acknowledgments mention two people, so some incongruency has apparently transpired, and we wonder about the reasons for that incongruency---perhaps the author accidentally used a graduate research assistant in a non-authorial capacity. The paper also declares (in line 96) an intention to “explore the impact of online digital art exhibitions of cultural heritage on visitors' expectations, emotional responses, and subsequent behavioral intentions,” but these intentions seem incredibly wide in scope, particularly with the reference to emotional responses. If the authors wanted to explore the notions of emotional responses, then some engagement with the disciplines of psychology, anthropology, cognitive sciences, and biology would have come about as a corollary to such a declared intention.
The paper has many ambiguous pronouns. In some examples among many others, in lines 683 to 685, the sentence "Firstly, while digital online exhibitions are rapidly developing worldwide, research on this topic is still in its infancy in China," the pronoun "its" might refer to the singular noun of "research" or the singular pronoun of "topic." In lines 139 to 142, the sentence "In the domain of online shopping, researchers such as Wang et al. (2011) have investigated the impact of visual stimuli on online shopping websites from the perspective of web aesthetics, exploring how consumers' psychological changes, in turn, influence their behavioral intentions in the online context [13]," the pronoun "their" can refer to the plural nouns of "intentions," "changes," "aesthetics," "websites," "stimuli," and "researchers," with no clear antecedent. The paper also abounds with examples of the eight passive verbs (am, are, be, being, been, is, was, were) that all scholarly authors should completely avoid.
Author Response
Point 1:
We unfortunately feel inclined to wonder about the appropriateness of this article’s inclusion (in the article’s current state) in the scholarly realm. While the author(s) probably rightly note the understudied nature of the so-called integration of online and offline platforms in the abstract, this paper serves as a reminder of the need for ingenuity and uniqueness during the discernment process that accompanies a researcher’s formulation of a research problem. Sometimes, a researcher decides to embark on a project that has all the appearances of appearing obviously or self-evidently inconsequential; even an infinite number of second glances would not transform those projects into matters of greater importance. Some researchers feel that these obviously inconsequential projects have never had any previous investigators, so such projects must naturally define viable research topics. The takeaway of this discussion lies in the fact that the mere novelty of a project does not turn that topic into a viable research project. The introduction also left this reviewer utterly confused as to what the researcher(s) wanted to demonstrate or prove, and this confusion probably had something to do with the opacity of the prose and the fact that the introduction seemed extremely discursive (i.e., jumping around so many topics in lengthy paragraphs whose topic sentences gave little clue over the content of the paragraphs).
In order to avoid the abovementioned issue(s), the author(s) should strongly consider a paper of an altogether different topic, namely an exploration of the nuances of how cultural heritage tourism administrators actually structure online exhibitions. In other words, one would want to know the reasons that animate individual cultural heritage tourism administrators to make certain selections regarding the intricacies of online exhibitions. We already know the basic reasons for the implementation of online exhibitions (i.e., a need to stay financially stable and to make cultural artifacts available for pandemic-restricted audiences, and so forth), but we wonder over the reasons that animate exhibitors to digitize one exhibition over another exhibition (i.e., a Song Dynasty-related piece, as the authors explain, over, say, a Tang-Dynasty piece). By delving into the mindsets and attitudes of museum administrators, the author(s) can write a far more illuminating paper devoid of the overly and unnecessarily abstruse theoretical jargon that impregnates a draft like this one---a draft that seems more like a sterile compilation of data than an actual attempt to analyze the data in interdisciplinary lenses that range from history to ethnography to cultural anthropology to other academic disciplines represented by museums. Such a paper would have a solid foundation in the disciplines of philosophy, history, anthropology, and archaeology---and the data of such a piece would serve as a handmaiden of the interpretive lenses of the author(s). As it now stands, the paper gives off an impression of wanting to simply stand on the data and little else. The current paper would probably have a more clearly defined and necessary role in the research and development committee of a government think tank, but not necessarily a scholarly journal.
Response to Reviewer:First and foremost, I would like to express my sincere gratitude for your genuine evaluation of this paper. Your feedback has provided us with better insights into the issues addressed in our research and has been immensely helpful to us. Based on your valuable suggestions, we have conducted profound revisions to the title, abstract, introduction, content, and conclusion of the paper. The specific modifications are as follows:
How has online digital technology influenced the on-site visitation behavior of tourists during the COVID-19 pandemic? A case study of online digital art exhibitions in China
Abstract: The COVID-19 pandemic has had a significant impact on the global tourism industry, leading to a decrease in peoples’ willingness to travel and a sense of insecurity regarding tourist destinations. Therefore, restoring people’s willingness to travel is the greatest challenge faced by this industry in the post-pandemic era. The tourism industry requires innovative solutions to achieve sustainable recovery. While there is a considerable amount of research on its recovery during the pandemic, there are few studies exploring people’s willingness to travel to encourage sustainable and resilient recovery in the post-pandemic era. This study employed a quality model to examine the satisfaction and intention of tourists towards the application of online digital art exhibitions under the influence of COVID-19. The aim was to investigate the promoting role of online digital art exhibitions in the sustainability and resilient recovery of the tourism industry. To achieve these objectives, this study focuses on the online digital art exhibition of Song Dynasty figure paintings launched by China Central Television (CCTV), with post-exhibition surveys conducted and 512 valid questionnaires collected. The research model and hypotheses are tested using structural equation modeling. The results of this study indicate that travelers' intentions to engage in on-site visits through online digital exhibitions are determined by three factors: perceived value, satisfaction, and art therapy. Furthermore, online digital art exhibitions not only represent the most important form of tourism during the pandemic, but also provide significant psychological healing. They have become a driving force for the transformation of the current culture and tourism industry and the promotion of its sustainable development. This research provides a benchmark for future research on the tourism industry, and offers new research directions in the field of sustainable tourism.
Keywords: sustainable tourism; tourism industry; COVID-19; behavioral intention; online digital exhibition
- Introduction
The tourism industry provides livelihoods to millions of people and enables billions to appreciate diverse cultures and the natural world. In some countries, tourism accounts for more than 20% of the gross domestic product (GDP), making it the third-largest export sector in the global economy [1]. However, the COVID-19 pandemic has severely impacted the tourism industry, affecting economies, livelihoods, and public services across continents and bringing the industry to a temporary halt [2]. According to forecasts by the World Travel & Tourism Council (WTTC), the COVID-19 pandemic was expected to cause a loss in global tourism value of USD 2.1 trillion in 2020 and jeopardize 75 million tourism jobs [3]. Rebuilding market confidence, revitalizing the tourism economy, and achieving high-quality development of the tourism industry have become focal points in academia and industry post-pandemic.
The outbreak of the pandemic has posed unprecedented challenges to the tourism industry, yet it has also presented an opportunity for transformation and change. The emergence of online digital technologies has facilitated the digital transformation and innovation of the tourism sector.[4]. Digital cultural tourism initiatives have emerged during the pandemic, leveraging new-generation technologies such as virtual reality (VR), augmented reality (AR), and artificial intelligence (AI) to create immersive experiences. These initiatives include virtual reality-based scenic spots, entertainment, and museums, and other new forms of culture and tourism experience, fostering new consumer behavior and reshaping people's travel preferences and tourism concepts [5]. Emphasizing safety in tourism and the ability to explore the world from home, and prioritizing user experience, have become paramount [5]. Under the circumstances of the pandemic, online digital technologies have gained more attention than ever from tourists and destination organizations. Online virtual tours of museums and exhibitions and other forms of "cloud tourism" have become the new norm since the pandemic, stimulating people's interest in tourism. Concurrently, online digital technologies are reshaping the manner in which customers plan their travel and search for destination information. By leveraging digitized environments, these technologies enable customers to experience products, services, or locations prior to their physical visit[6]. The increasing reliance on and desire for such online digital environments reflects a gradual shift in tourism perspectives, with individuals predominantly opting for online virtual tourism. Consequently, online digital technologies have opened up new avenues for the revitalization and recovery of the tourism industry, providing innovative approaches to travel and tourism experiences.
Previous studies have explored the use of on-site digital technologies [7] and VR technology for enhancing tourists' travel experiences [8], as well as the impact of VR technology on intentions to make repeat visits to tourist destinations [9-10]. However, the perspectives explored in these studies have been relatively narrow, focusing solely on the impact of digital technologies on tourism destinations. They predominantly concentrate on the "technology" itself, without delving into the influence of online digital technologies on "people" after the pandemic. Particularly, the shift in individuals' tourism perspectives and their heightened concerns for personal well-being and health have significantly dampened their willingness to travel. Yet, the revival of tourist demand is considered a key factor in stimulating the recovery of the tourism industry[11] . Consequently, there is an urgent need to address how to effectively guide tourists in regaining their willingness to travel. In fact, tourists are highly interested in online digital technologies and seek to understand how these platforms can provide humanistic care[12-13]. However, there have been few attempts to evaluate user satisfaction with online digital technology applications and their subsequent impact on travel intentions. How do online digital technologies influence tourists' on-site visitation behavior during the COVID-19 pandemic? What are the underlying factors affecting this behavior? Do they contribute to stimulating tourists' travel intentions and promoting the recovery and sustainable development of the tourism industry in the post-pandemic era?
To address the aforementioned issues, this study introduces a comprehensive research model that incorporates factors derived from perceived quality (content, system, and personalized service) and expectation confirmation. From the perspective of tourists' expectation confirmation, the research model explores the effects of perceived quality of online digital art exhibitions, user expectation confirmation, perceived value, perceived enjoyment, and art therapy on satisfaction with online digital art exhibitions and on-site visitation. By examining how online digital technology influences the physical visit behavior of tourists during the COVID-19 pandemic, this research can assist museum curators and other cultural tourism institutions in leveraging digital technology to increase public engagement with art, promote the digital transformation of the culture and tourism industry, stimulate innovation in this sector, unleash the multiplicative effects of digitization on the industry, and facilitate its sustainable development.
2.Research Theory and Hypothesis
2.1. COVID-19 and online digital art Exhibitions
In the spring of 2020, as the novel coronavirus began to spread globally, the true meaning of globalization seemed to resonate with people, as the impact of the virus extended beyond trade disruptions and stagnant tourism to endanger the art world [14]. As a result of the pandemic, planned art fairs and major exhibitions in various countries were hindered, while museums in many countries remained closed, presenting challenges in cross-border artistic collaboration [15]. To cater to audiences who were unable to attend in person, many museums swiftly transitioned their exhibitions to online platforms, offering "virtual exhibitions" for remote viewing. Suddenly, being "online" became the hottest topic of discussion. It is during this distinctive period that the rapid development of digital art took place.
Online digital art exhibitions refer to the integration of online digital technologies with humanistic art, combining rational thinking and artistic sensibility. These exhibitions are founded on the development of online digital technologies and encompass a fusion of artistic expression and human perception.[16]. Online exhibitions, as a form of exhibition accessible anytime and from anywhere through the internet on computers and smartphones, represent one of the most effective ways to disseminate digital information in any field [17]. Online digital art exhibitions represent a new artistic language that blends the contemporary era of new media with traditional art. Leveraging technology for human–computer interaction as a medium, these exhibitions enhance visitors' perceptual experiences and create a multidimensional and dynamic interactive environment.
- Discussion
5.1. Discussion
The global tourism industry has been severely impacted by the COVID-19 pandemic, which is ongoing and becoming the new norm. As the world gradually recovers, stakeholders in the tourism industry are seeking policies that can facilitate its revival in this new context. Therefore, one of the primary objectives of this study is to assess the impact of online digital art exhibitions on the recovery and revitalization of the tourism industry. This study analyzes the decisive factors in visitors' on-site visitation behavior as influenced by the widespread adoption of online digital technologies during the COVID-19 pandemic. The findings, presented in Table 5, provide support for all ten of our proposed hypotheses.
This study demonstrates that the three dimensions of perceived quality in online digital art exhibitions (content quality, system quality, and service quality) are crucial in confirming visitors' expectations, with service quality having the highest impact coefficient. These findings are consistent with those of previous studies [79-81] . Specifically, the more stimuli visitors experience from online digital art exhibitions on cultural heritage, the more positive their confirmation becomes. The iconic representation of Song Dynasty figure paintings in Chinese aesthetics serves as an attraction for visitors, as the perceived visual aesthetic appeal and diverse interactive experiences stimulate higher levels of expectation confirmation. Therefore, as people's travel demands evolve, future online curators should consider leveraging the increased cognitive recognition of this type of exhibition tourism, focusing on broader sharing, more effective interactions, richer sensory experiences, and convenient information dissemination.
Secondly, the empirical results of this study reveal the existence of two mediating effects between expectation confirmation and satisfaction, which indirectly influence visitors' intentions to visit offline museums. These effects include the mediation of perceived value and enjoyment, as well as enjoyment and art therapy. This implies that perceived value, enjoyment, and art therapy are important pathways through which expectation confirmation influences satisfaction. Expectation confirmation serves as a significant motivating factor for satisfaction, and while previous studies have confirmed the impact of expectation confirmation on satisfaction [82-83], the internal process mechanism underlying the influence of expectation confirmation on satisfaction remains unclear. This study discovers that perceived value, enjoyment, and art therapy can further elucidate the relationship between expectation confirmation and satisfaction. Perceived value plays a partially mediating role in confirmation and satisfaction, which is consistent with previous research demonstrating the important link between perceived value and expectation confirmation and satisfaction [84]. Additionally, for the first time, this study reveals that expectation confirmation influences satisfaction through the mediation of enjoyment and art therapy, with remote mediation effects being significant. In other words, the higher the expectation confirmation of visitors to online cultural heritage art exhibitions, the higher their perceived enjoyment, and the greater their perception of art therapy, leading to physical relaxation and mental comfort; this, in turn, affects visitors’ satisfaction with online cultural heritage digital art exhibitions. These findings indicate that the importance of expectation confirmation in ensuring satisfaction can also be understood through the mediating roles of enjoyment and art therapy.
Thirdly, this study reveals that perceived value, satisfaction, and art therapy, as latent variables, exert a positive and significant influence on visitors' offline visitation behavior. These findings align with those of previous research [84], confirming the consistent impact of perceived value and satisfaction on visitors' offline visitation behavior. Moreover, this study provides support for the notion that there is a greater influence of perceived value on behavioral intention compared to satisfaction [85]. Specifically, visitors perceive online digital art exhibitions as possessing high literary and historical value, making them worth visiting. Visitors’ behavioral intentions are more strongly driven by perceived value than satisfaction. Additionally, the inclusion of art therapy as a predictor of visitors' on-site visitation behavior represents a novel discovery in this study. Previous research suggests that experiencing a healing sensation from a favorable environmental stimulus strongly affects visitors' attachment to a place and behavioral intentions [86]. This indicates that the confirmation of visitors’ emotions arising from the art therapy effect of online cultural heritage digital art exhibitions can effectively predict their intention to engage in on-site visitation and their attachment to the art exhibition. Although COVID-19 is gradually receding, the psychological trauma caused by the pandemic lingers and cannot be easily alleviated. However, through immersive online experiences that allow visitors to immerse themselves in Song and Yuan aesthetics, appreciate the daily culture of the Song Dynasty spanning thousands of years, interact with the exhibits, and learn new information, individuals can effectively address emotions such as anxiety, depression, and anger, thereby promoting psychological healing. Consequently, visitors are more inclined to visit offline museums and personally experience the spiritual healing that art offers.
6.Conclusions and Suggestions
6.1.Theoretical implications
Firstly, this study establishes a new S-O-R model based on the expectation confirmation model, perceived quality, and art therapy theory. The model explains 48% of the variance in visitors' behavioral intentions to engage in on-site visitation, providing a cognitive framework to analyze the logic underlying the interaction between the perceived quality of online digital art exhibitions and on-site visitation behavior. It delves into the stimuli of perceived quality experienced by visitors through online digital art exhibitions, leading to perceptions of increased value, enjoyment, and art therapy, thereby influencing visitors' decisions to engage in physical visits. Therefore, the model developed in this study serves as empirical evidence of the benefits of using digital technology in tourism experiences and the communication of cultural heritage tourism.
Secondly, we offer an effective method to assess whether online digital art exhibitions meet visitors' expectations regarding enjoyment, value, and art therapy, thereby enhancing their likelihood of engaging in on-site visitation behavior. These research findings will enable cultural heritage tourism managers to understand visitors' motivations, allowing them to design online digital art exhibitions that cater to visitors' psychological needs and encourage them to visit on-site. This is particularly significant in the aftermath of the COVID-19 pandemic. In fact, online digital art exhibitions provide an alternative for individuals who may be interested in art exhibition content but are unable to physically attend. Furthermore, our research results may encourage web designers of online digital art exhibitions to improve the usability and aesthetics of their online interfaces. By effectively utilizing the Internet, multimedia, emotional content, and aesthetic interfaces, online digital art exhibitions can attract a larger audience and generate interest in art exhibitions and visiting environments within the cultural domain. In the digital information age, online cultural heritage art exhibitions bear the social mission of informing the public about historical civilizations and preserving cultural heritage.
6.2. Practical Inspiration
Firstly, this study could provide practical insights for tourism managers, as online digital cultural tourism is a key driver in the transformation of the culture and tourism industry. With the accelerated process of digitization, blockchain-based digital art has become a new trend in the wave of digital economic development [87]. The "digital + culture" development model presents infinite possibilities for the revitalization of China's remarkable culture and the development of cultural tourism. Digital transformation has promoted the transformation and upgrading of the tourism industry. Online digital art exhibitions, representing a transition from traditional offline exhibition formats to intelligent online travel experiences, are propelling the tourism industry towards greater efficiency and intelligence.
Secondly, this study provides guidance for tourism managers on the bidirectional integration of online and offline exhibition technologies. Online exhibitions focus on personalized services, offering visitors the opportunity to access and enjoy cultural and artistic activities anytime, while reducing the use of travel resources and associated environmental pressures such as carbon emissions from air travel. It significantly reduces the energy costs associated with maintaining buildings, hosting exhibitions, and travel. This is particularly beneficial for those who have limited mobility or lack economic resources for travel, thereby promoting the sustainability of the tourism industry. On the other hand, offline exhibitions emphasize quality and provide unique tourism value through interactive settings and real-time experiences. Visitor engagement in cultural and artistic activities can stimulate the local community's economic and cultural life. The integration of online and offline experiences mutually reinforces the sustainability of the tourism industry.
Thirdly, this study contributes to the understanding of how online digital technologies can revitalize historical and cultural heritage in the field of culture and tourism management. Online digital exhibitions allow developers to vividly present information on historical and cultural heritage and folk culture, urban stories, and brand concepts to visitors. This not only promotes the vitality of traditional culture and tourism industry values and commercial values, but also satisfies the public's curiosity and thirst for knowledge through interactive experiences using digital technology. Thus, cultural heritage is effectively protected while being showcased and utilized, providing more inspiration and forms of expression for cultural heritage inheritance and development.
Furthermore, this research contributes to enhancing citizens' well-being. In the context of the post-pandemic era, online digital art exhibitions can serve as healing spaces, alleviating the pressures of temporary living, working, and studying, and nurturing psychological well-being. In this sense, online digital art exhibitions deserve attention from curators and designers. Unrestricted by space and distance, online digital art exhibitions have a large audience and wide dissemination, making them the fastest and most effective platforms for art therapy. Today's online digital art exhibitions extend beyond simply replicating offline museum and art gallery experiences. Through the integration of technology and art, they emphasize individual spiritual pleasure and freedom. Cognitive psychology also demonstrates that high-quality interactive art is more likely to enable viewers to contemplate their own existence, experience basic sensory and behavioral responses, and awaken deep emotional resonance [88]. By guiding audiences from the real world to the virtual world, a dialogue between individuals and artworks is accomplished, incorporating art exhibitions into users' lifestyles and elevating the overall spiritual and cultural development of society.
6.3. Limitations and Future Work
This study has several limitations that should be acknowledged. Firstly, digital technology tourism is developing rapidly worldwide, but research on this topic in China is still in its early stages. Additionally, the sample used in this study may lack diversity across ethnic and cultural backgrounds, as visitor behavior may be influenced by their ethnic culture. Therefore, the generalizability of our research findings needs to be extended to other countries with well-developed online digital exhibitions and different ethnic cultures. Secondly, due to time constraints, we relied on cross-sectional experiential data to measure customer perceptions and behavior. The internal changes that occurred among visitors remain unknown. Considering the ongoing evolution of online digital exhibition content, features, and services, as well as the advancements in technology, measurement biases resulting from time gaps may be increased. Hence, we recommend that future researchers, given sufficient funding and time, collect longitudinal empirical data to examine the interactions among different temporal variables, thereby obtaining more effective and robust validation results.
Point 2:
For some miscellaneous remarks, we should like to add the following. The paper oddly ends at line 694 in a sentence that does not terminate with a punctuation mark, suggesting that the authors wanted to go on with some further discussion that never quite materialized.
Response to Reviewer:Thank you for your valuable suggestions. The authors of this study have carefully reviewed the content at line 694 and identified missing and confusing parts due to translation issues. As a result, the paper underwent further revisions and was thoroughly reviewed. Finally, professional English language experts were consulted to ensure the accuracy of the content in its final form.
Point 3: In addition, the paper indicates a single author, but the acknowledgments mention two people, so some incongruency has apparently transpired, and we wonder about the reasons for that incongruency---perhaps the author accidentally used a graduate research assistant in a non-authorial capacity.
Response to Reviewer:Thank you sincerely for your genuine evaluation of the paper. I would like to address the mention of a fellow student from the research laboratory in the acknowledgments section. Although they provided limited guidance on using the Amos software during my data analysis, their contribution to the actual writing of the paper was not substantial. I included their mention out of courtesy. Furthermore, I express my gratitude to my university for providing a conducive research platform and financial support, which facilitated the smooth progress of our study. Once again, I am deeply grateful for your meticulous suggestions and approach towards the paper. They have greatly benefited us as authors and provided invaluable assistance in the process of revising our paper.
Point4: The paper has many ambiguous pronouns. In some examples among many others, in lines 683 to 685, the sentence "Firstly, while digital online exhibitions are rapidly developing worldwide, research on this topic is still in its infancy in China," the pronoun "its" might refer to the singular noun of "research" or the singular pronoun of "topic." In lines 139 to 142, the sentence "In the domain of online shopping, researchers such as Wang et al. (2011) have investigated the impact of visual stimuli on online shopping websites from the perspective of web aesthetics, exploring how consumers' psychological changes, in turn, influence their behavioral intentions in the online context [13]," the pronoun "their" can refer to the plural nouns of "intentions," "changes," "aesthetics," "websites," "stimuli," and "researchers," with no clear antecedent. The paper also abounds with examples of the eight passive verbs (am, are, be, being, been, is, was, were) that all scholarly authors should completely avoid.
Response to Reviewer:Thank you for your meticulous approach to the paper. The authors of this study have engaged professional English translators for proofreading after completing the paper's modifications, ensuring that the final version is comprehensible.

With best regards
Response to Reviewer:Thanks for your recognition and sincere comments. With your help, we have a more quality paper, and your comments inspired us greatly. Again, thanks for your kindness.

Round 2
Reviewer 3 Report
The revisions greatly improved the article. The only thing that remains are a few typos.
Reviewer 5 Report
The researchers addressed the broad scope of our feedback and incorporated our suggestions.
Grammatical issues on ambiguous pronouns still abound in the paper, i.e. (among many others)
Abstract
While there is a considerable amount of research on its recovery during ing the pandemic, there are few studies exploring people’s willingness to travel to encourage sustainable and resilient recovery in the post-pandemic era.
(No idea about the antecedent for "its")
They have become a driving force for the transformation of the current culture and tourism industry and the promotion of its sustainable development.
(No idea about the antecedent for "its" = promotion(?), industry(?), culture(?0, transformation(?), force(?) = one, and only absolutely only one, singular noun should precede the possessive pronoun "its" in the same sentence)
Lines 89~93
By examining how online digital technology influences the physical visit behavior of tourists during the COVID-19 pandemic, this research can assist museum curators and other cultural tourism institutions in leveraging digital technology to increase public engagement with art, promote the digital transformation of the culture and tourism industry, stimulate innovation in this sector, unleash the multiplicative effects of digitization on the industry, and facilitate its sustainable development.
Its = industry, digitization, sector, innovation, art, technology, engagement (?)